# What If the Input is Expanded in OOD Detection?

**Boxuan Zhang**[1][*]    **Jianing Zhu**[2][*]    **Zengmao Wang**[1][†]    **Tongliang Liu**[3]    **Bo Du**[1]    **Bo Han**[2,4]

[1]School of Computer Science, Wuhan University
[2]TMLR Group, Department of Computer Science, Hong Kong Baptist University
[3]Sydney AI Center, The University of Sydney    [4]RIKEN Center for Advanced Intelligence Project

{boxzhang1005, wangzengmao, dubo}@whu.edu.cn
{csjnzhu, bhanml}@comp.hkbu.edu.hk    tongliang.liu@sydney.edu.cn

## Abstract

Out-of-distribution (OOD) detection aims to identify OOD inputs from unknown classes, which is important for the reliable deployment of machine learning models in the open world. Various scoring functions are proposed to distinguish it from in-distribution (ID) data. However, existing methods generally focus on excavating the discriminative information from a single input, which implicitly limits its representation dimension. In this work, we introduce a novel perspective, i.e., employing different common corruptions on the input space, to expand that. We reveal an interesting phenomenon termed *confidence mutation*, where the confidence of OOD data can decrease significantly under the corruptions, while the ID data shows a higher confidence expectation considering the resistance of semantic features. Based on that, we formalize a new scoring method, namely, ***Confidence aVerage*** (CoVer), which can capture the dynamic differences by simply averaging the scores obtained from different corrupted inputs and the original ones, making the OOD and ID distributions more separable in detection tasks. Extensive experiments and analyses have been conducted to understand and verify the effectiveness of CoVer. The code is publicly available at: https://github.com/tmlr-group/CoVer.

## 1 Introduction

Out-of-distribution (OOD) detection [23, 28, 44] is important for reliable machine learning model deployment in open-world scenarios, where various samples from unknown classes, i.e., OOD data, are constantly emerging [4]. Deep neural networks [20] (DNNs) are demonstrated to be overconfident about these OOD data, which may result in disasters for some safety-critical applications [5, 21]. Traditional OOD detection methods [23, 29, 28, 30, 45, 46, 1, 61] design various scoring functions based on the outputs or representations extracted from well-trained models. Recently, some research also extended it into a zero-shot setting [31], which leverages the multi-modal information based on vision-language models (VLMs) and requires no further training on in-distribution (ID) data. A series of methods [48, 36, 52, 26] are proposed for improving OOD detection based on such advances.

Although promising progress has been achieved, existing methods mainly focus on excavating the discriminative information of a single input. For instance, ReAct [45], DICE [46], and ASH [1] integrates the activation regularization or reshaping to the forward path of a single input in single-modal DNNs; MCM [31] characterizes the confidence of a single input by the similarity between visual features and text representation of ID classes in VLMs. However, specializing in a single input

---
[*]Equal Contribution
[†]Correspondence to Zengmao Wang (wangzengmao@whu.edu.cn)

38th Conference on Neural Information Processing Systems (NeurIPS 2024).

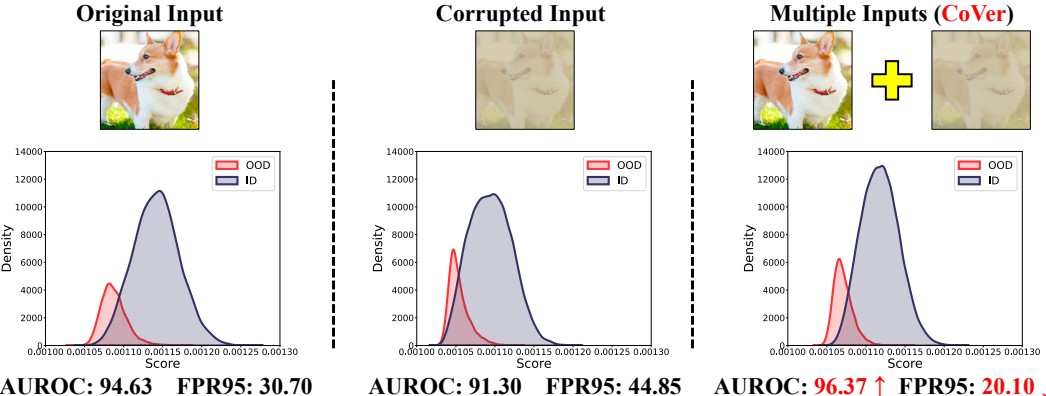

Figure 1: Comparison of scores distributions and detection results with different inputs for representation dimension expansion. *Left panel*: results with a single original input; *Middle panel*: results with a single corrupted input, which perform worse but have mutated scores for some OOD samples (see Figure 2); *Right panel*: results with multiple inputs (CoVer), which achieve the variance reduction for the ID distribution and perform a better ID-OOD separability (see Figure 3 for more explanations).

may implicitly constrain the representation dimension for detection, leaving some hard-to-distinguish OOD samples with features similar to ID samples fail to be identified (refer to the distribution overlap in the left panel of Figure 1). Therefore, it naturally motivates the following critical research question: *Can we expand the dimension of the input space to explore OOD discriminative representations?*

In this work, we introduce a novel perspective to investigate that, i.e., employing the common corruptions [22] in the input space. Through a systematical comparison, we reveal an interesting phenomenon termed *confidence mutation*, where the confidence of OOD data can decrease significantly under the corruptions, while ID data shows higher confidence expectation considering different input dimensions. Specifically, as shown in Figure 1, corrupted inputs result in lower confidence in both OOD and ID data. However, one critical dynamic discovery is that its confidence about overconfident OOD data is changed more than the unconfident ID data under the same corruptions (refer to Figure 2), indicating a natural difference in feature-level resistance of the originally overlapped parts (refer to Figure 3). With the original inputs, we can find that the model is overall more confident in ID data.

Based on the above, we propose a new scoring framework, namely, *Confidence aVerage* (CoVer), as illustrated in Figure 4. At the high level, we expand the original single representation dimension into multiple ones to excavate discriminative information. In detail, we introduce a simple but effective method for identifying OOD data with confidence mutation, which can be formalized as an average of OOD scores (e.g., Eq. (6)) obtained by different corrupted inputs and the original one. With the expectation among multiple input dimensions, CoVer can effectively reflect the knowledge of invariant semantic features that are discriminative from ID data to OOD data. It also matches an intuition that ID data can be more likely recognized as high confidence by models considering different input views, especially with the corruptions affecting the non-semantic high-frequency parts.

We conducted extensive experiments to verify the effectiveness of our proposed method. Since CoVer is an input-side design compatible with single-modal and multi-modal networks, we adpot various benchmarks for DNN-based and VLM-based OOD detection tasks. Under extensive evaluations, our CoVer can achieve the superior performance compared with different baselines. Moreover, CoVer exhibits excellent compatibility, as evidenced by the better performance of some methods combined with CoVer. Finally, a range of ablation studies of the scoring framework and further discussions from different perspectives are provided. In summary, our main contributions can be listed as follows,

- Conceptually, we introduce a novel perspective for identifying OOD inputs by considering the common corruptions to expand the representation dimensions. (in Section 3.1)

- We reveal an interesting phenomenon termed *confidence mutation*, where the confidences of OOD data can vary to significantly lower than ID data under corruptions (in Section 3.2).

- Technically, we formalize a novel scoring method, namely, *Confidence aVerage* (CoVer), a simple average of the confidence estimated from extended corrupted inputs and the original one. The corresponding empirical analysis is presented to understand it (in Section 3.3).

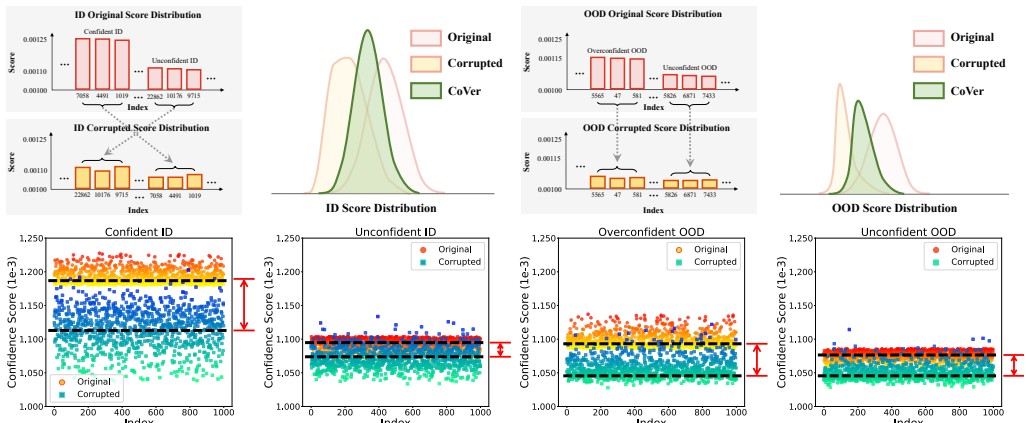

Figure 2: Demonstration about detailed explanations for the discovery illustrated in Figure 1. The ID and OOD data here are divided into four groups, i.e., Confident ID, Unconfident ID, Overconfident OOD, and Unconfident OOD. *First Row*: the variation of confidence scores for ID and OOD data before and after being corrupted. The critical difference lies in the greater confidence declination for overconfident OOD data compared to unconfident ID data. (see Figure 3 for further discussion). *Second Row*: scatter maps of confidence scores sampled from the four groups under the same corruption, statistically supporting the findings of the first row. See Appendix C.2 for more details.

- Empirically, extensive experiments on both traditional and zero-shot OOD detection benchmarks have verified the effectiveness and compatibility of our CoVer, and we conduct various ablations and further discussions to provide a comprehensive analysis (in Section 4).

## 2 Preliminaries

In this section, we briefly introduce the preliminaries of OOD detection on basic setups and the advanced zero-shot setting on VLMs. For related works, we leave detailed discussions in Appendix B.

**Problem setups.** Let $\mathcal{X}$ be the input space and $\mathcal{Y} = \{y_1, ..., y_K\}$ be the label space, where $K$ is the number of ID classes. Given the ID random input $x_i \in \mathcal{X}$ and OOD random input $x_o \in \mathcal{X}$, we consider the ID marginal distribution $\mathcal{D}_{\text{ID}}$ and those with the OOD marginal distribution $\mathcal{D}_{\text{OOD}}$, where $\mathcal{D}_{\text{OOD}}$ is defined as an irrelevant distribution of which the label set has no intersection with $\mathcal{Y}$. The goal of OOD detection is to figure out inputs with the OOD distribution $\mathcal{D}_{\text{ID}}$ from those with the ID distribution $\mathcal{D}_{\text{OOD}}$, which can be considered as a binary classification problem. For traditional OOD detection, given a model $f$ trained on ID data with logit outputs, a score function $S(\cdot)$ and a threshold $\lambda$, the detection model $g(\cdot)$ can be defined as,

$$g_\lambda(x) = \text{ID}, \quad \text{If } S(x; f) \geq \lambda; \text{ otherwise}, \quad g_\lambda(x) = \text{OOD}. \quad (1)$$

where $x$ is detected as ID data if and only if $S(\mathbf{x}) \geq \lambda$; otherwise, it is rejected as OOD data that should not be predicted by the model $f$. Designing a practical $S(x; f)$ is crucial for OOD detection.

**CLIP-based vision-language models** CLIP [42] has shown impressive performance in the zero-shot classification task by profiting from massive amounts of training data and large-size models. Here we briefly review the mechanism of CLIP-based VLMs. A CLIP-based model $\mathbf{f}$ usually contains an image encoder $\mathbf{f}^{\text{image}}$ and a text encoder $\mathbf{f}^{\text{text}}$. Given a random input $x \sim \mathcal{D}_{\text{ID}}$ and a label $y \sim \mathcal{Y}$, we use $\mathbf{f}^{\text{image}}$ and $\mathbf{f}^{\text{text}}$ to extract the image features $h \in \mathbb{R}^d$ and the text features $e_j \in \mathbb{R}^d$ as follows:

$$h = \mathbf{f}^{\text{image}}(x), \quad e_j = \mathbf{f}^{\text{text}}(p(y_j)), \quad \forall j = 1, 2, ..., K, \quad (2)$$

where $p(\cdot)$ refers to the prompt template for the input label, $d$ is the embedding dimension. The predictions are formulated as the consine similarity between the image features $h$ and text features $e_j$,

$$\hat{y} = \underset{y_j \in \mathcal{Y}}{\arg \max} \{\cos(h, e_j)\}, \quad \text{where} \quad e_j = \mathbf{f}^{\text{text}}(p(y_j)). \quad (3)$$

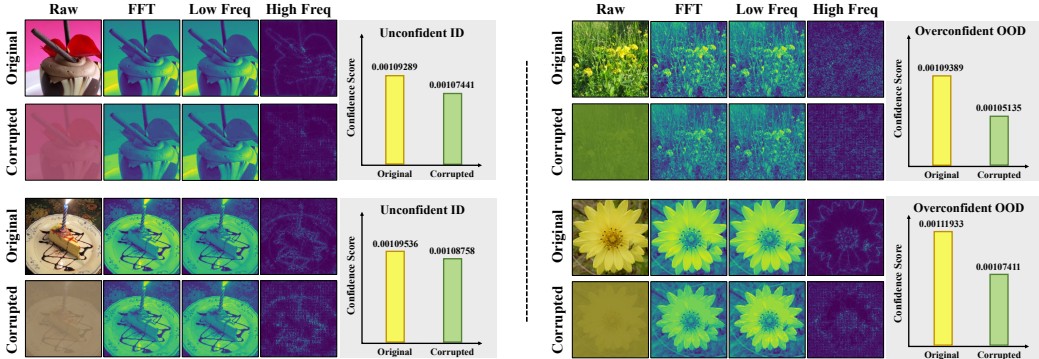

Figure 3: Visual exploration of random unconfident ID samples and the confidence mutation exemplified on random overconfident OOD samples under the same corruption. For each original input and its corrupted variant, we leverage the Fast Fourier Transformation to extract their low-frequency and high-frequency parts. *Left panel*: visual investigation on unconfident ID samples with ID semantic features at low-frequency levels that are resistant to corruptions. *Right panel*: an intuitive comparison of overconfident OOD samples, whose confidences show significant changes due to the elimination of non-semantic features at the high-frequency level. See Appendix C.4.2 for more detailed analyses.

**Zero-shot OOD detection**    Different from traditional OOD detection methods based on a classifier $f$ well-trained on single-modal, recent zero-shot OOD detection studies [31, 26] leverage a pre-trained VLM-based model (e.g. CLIP [42]) without any fine-tuning on ID training data. The text features from given ID label names (i.e. ID classes) as the class-wise weights functionally play the same role as the classifier. With guaranteed ID classification accuracy, the primary goal of zero-shot OOD detection in this paper is to distinguish OOD samples that do not belong to any known ID classes.

## 3    CoVer: Confidence Average

In this section, we formally present our proposed new scoring framework, i.e., *Confidence aVerage* (CoVer). First, we introduce the motivation of representation dimension expansion and present the notable discovery (Section 3.1). Second, we conduct the exploration for the confidence mutation of overconfident OOD data considering the corrupted inputs (Section 3.2). Lastly, we provide the detailed implementation and analysis of our formalized CoVer score (Section 3.3).

### 3.1    Representation Dimension Expansion

DNNs are demonstrated to be overconfident on those OOD samples, and a series of works [23, 45, 1, 54, 31, 26] are dedicated to eliminating the effects through the perspective of feature representation. However, achieving that is demonstrated to be hard as it generally requires careful optimization [54], or additional prior knowledge [36] on the single input. As illustrated in the middle panel of Figure 1, adopting some agnostic corruptions on the single input may result in worse separability between the ID and OOD distribution. Specializing in a single input seems to implicitly constrain the representation dimension for detection. In this work, we naturally raise the following question,

*What if we expand the dimension of representation for the original inputs*
*to enhance OOD discriminative representations?*

Using the same corruption adopted in Figure 1, we can conduct the dimension expansion by simultaneously considering both the corrupted variant and the original input. One notable discovery is that considering multiple inputs can achieve better performance on OOD detection, even though the single corruption transformation brings negative effects on identifying OOD samples. The surprising comparison results attract us to further explore the underlying mechanism of expanding the representation dimension for the original input, especially the dynamics before and after adopting the corruptions.

## 3.2 Confidence Mutation under Corruptions

Although employing corruption on the single inputs leads to worse ID-OOD separability, we can find obvious shifts toward less confidence in both OOD and ID distributions. It is expected for OOD data that corruption can help the model mitigate overconfidence, while the ID data are also affected severely and enlarge the overlap on the single dimension. In contrast, considering the multiple inputs by averaging the confidence scores shows variance reduction for ID distribution, which indicates the distinguishable dynamics between ID and OOD data. To elicit the underlying mechanism, we provide a definition to characterize the change of model confidence on the inputs under corruption.

**Definition 3.1** (Confidence Difference). *Given a well-trained model $f$ and a score function $S(\cdot)$ measuring the confidence of $f$ on an input $x$, we have a basic static to characterize the differences between the original input and that under a corruption $c(\cdot)$: $\mathrm{MU}_c(x, S, f) \triangleq (S(x; f) - S(c(x); f))$.*

Based on the comparison in Figure 1, we divide the ID and OOD data into four groups according to the model confidence on their original inputs, and present an overall comparison of the confidence differences in Figure 2. We reveal the critical dynamic differences in the corrupted variants of ID and OOD data, where both large $\mathrm{MU}(x, s, f)$ exist in the data whose natural inputs own higher confidence in each part, demonstrating the model confidence on the overconfident OOD data decrease more than the unconfident ID data under the same corruption. We can get the empirical observation,

**Observation 3.2** (Confidence Mutation). *Given the overconfidence OOD inputs $x_o \in \mathcal{D}_{OOD}$, we can observe more significant differences in the change of confidences under the same corruption $c(\cdot)$ than the ID samples $x_i \in \mathcal{D}_{ID}$ with similar model confidence (constrained by $\epsilon$) on the natural inputs,*

$$\mathbb{E}_{x_i \sim \mathcal{D}_{\mathrm{ID}}}(\mathrm{MU}_c(x_i, S, f)) < \mathbb{E}_{x_o \sim \mathcal{D}_{\mathrm{OOD}}}(\mathrm{MU}_c(x_o, S, f)). \tag{4}$$

In Figure 3, we further visualize the samples of unconfident ID data and overconfident OOD data. Under the comparison of saliency maps and the Fast Fourier Transformation, we find the confidence mutation reflects the feature level vulnerability of OOD data compared with ID data. Intuitively, the former can be severely affected by the common corruption to eliminate the non-semantic features, which generally exist at the high-frequency level. In contrast, the semantic feature of unconfident ID data can maintain confidence as the limited effects of corruption on the low-frequency part.

**Observation 3.3** (Resistance of ID features in frequency views). *Assuming that ID data owns the ID semantic features existing at the low-frequency level (extract by $\Gamma_\xi$) while the OOD data has some non-semantic features at the high-frequency level for activating the high confidence of the model on its prediction, we can observe the following empirical relation on adopting the same corruptions,*

$$\mathbb{E}(\mathrm{MU}_c(x, S, f)) \propto \mathrm{KL}((f(\Gamma_\xi(c(x)))) || f(\Gamma_\xi(x)). \tag{5}$$

where $\Gamma$ indicates the Fourier transformation. We suggest that common corruptions [22] might act as perturbations of high-frequency features within the input representation. For OOD samples, which inherently lack ID semantic features, altering high-frequency features could potentially lead to notable changes in model confidence, while the ID data shows relatively better resistance on it (see the left panel of Figure 3). This observation tentatively supports the notion that ID data maintains an overall higher confidence expectation under conditions of expanded representation dimension. To validate its generality, additional results involving various common corruptions are presented in Appendix C.2.2.

## 3.3 Scoring Function Implementation and Analysis

Based on the previous understanding of confidence mutation, we formalize our CoVer, a new scoring framework for OOD detection. The procedure of CoVer mainly contains the following *four* parts as illustrated in Figure 4, and the final averaged multi-dimensional scores can be provided as follows,

$$S_{\mathrm{CoVer}} = \mathbb{E}_{x \sim d(\mathcal{X}, \tilde{\mathcal{X}})} \max_i \frac{e^{s_i(x)/\tau}}{\sum_{j=1}^K e^{s_j(x)/\tau}}, \quad d(\mathcal{X}, \tilde{\mathcal{X}}) := \{x, c(x) | x \in \mathcal{X}, c \in C\}, \tag{6}$$

where $\mathbb{E}_{x \sim d(\mathcal{X}, \tilde{\mathcal{X}})}$ is the confidence expecation over all scores dimensions, $K$ is the number of ID classes, $\tau$ is the temperature coefficient of the softmax function. In the following, we detailedly introduce the specific operations to obtain the final $S_{\mathrm{CoVer}}$ and the corresponding notations.

To enlarge the dimension of the original single input for confidence average, we introduce various corrupted inputs. In this work, we employ the corruption functions defined in [22], which consists

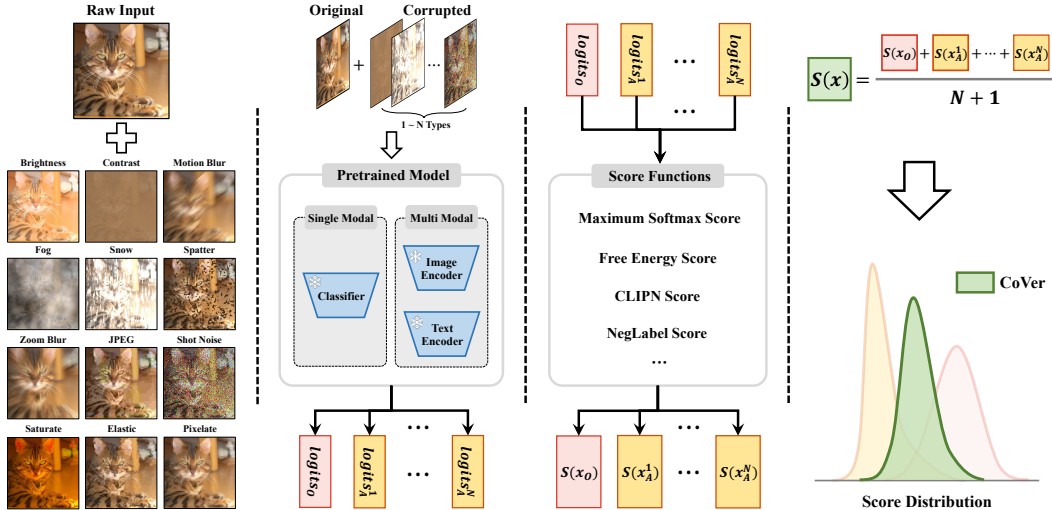

Figure 4: Overview of CoVer. *Left panel*: visualization of the raw input and inputs w.r.t different corruptions; *Left-middle panel*: procedures of logit outputs from single-modal and multi-modal networks; *Right-middle panel*: scoring functions that equip each dimensional output with an OOD score; *Right panel*: realization of CoVer by averaging OOD scores obtained from multiple dimensions.

total of 90 distinct corruptions. We provide the visualization of these different corruptions in Appendix C.4.1. Given the input space $\mathcal{X}$ and a set of corruption functions $C$, the corrupted inputs can be formulated as $\{c(x)|x \in \mathcal{X}, c \in C\} \rightarrow \tilde{\mathcal{X}}$, resulting in the multi-dimensional input spaces $d(\mathcal{X}, \tilde{\mathcal{X}})$.

Given an input image $x \sim d(\mathcal{X}, \tilde{\mathcal{X}})$, we adopt an image encoder with fixed parameters to extract the feature of the original dimension $h_O$ and features of corrupted dimensions $h_1, ...h_N$. Then we predict the logit output $s(x)$ for each dimensional feature $h_d, \forall d = O, 1, ..., N$. For the DNN-based model $f$, the outputs of these features are denoted as $s(x) = f(x) = \text{logits}_d$. For the VLM-based model, the outputs are label-wise matching scores based on the cosine similarity: $s_j(x) = \frac{h_d \cdot e_j}{\|h_d\| \cdot \|e_j\|}$.

For the logit output $s(x)$ predicted from a specific input dimension, we assign it with an OOD score to implement one dimension of the CoVer score. As shown in the right-middle panel of Figure 1, the OOD score can be formalized by some traditional scoring functions, like the softmax scoring function [23] (refer to Eq. (6)) and the free energy scoring function [30]. In addition, the OOD score can also be formulated by variants of some novel scoring functions, like those in CLIPN [52] and NegLabel [26]. The detailed implementations for alternative scoring functions can be found in Appendix C.3.

## 4 Experiments

In this section, we present the comprehensive verification of the proposed CoVer in the OOD detection benchmarks. First, we introduce several critical parts of experimental setups (in Section 4.1). Second, we provide the performance comparison and compatibility experiments of our CoVer with various DNN-based and VLM-based OOD detection methods (in Section 4.2). Third, we conduct extensive ablation studies and further discussions to understand the properties of our CoVer (in Section 4.3).

### 4.1 Experimental Setups

In this part, we present the critical parts of experimental setups and leave more details in Appendix C.

**Datasets.** Following previous work [1, 31], we adopt the ImageNet-1K OOD benchmark [24], which uses the ImageNet-1K [14] as ID data and iNaturalist [49], SUN [55], Places [60], and Textures [7] as OOD data. For each of the OOD datasets, the classes do not overlap with the ID dataset. As the same as MCM [31], we also use subsets of ImageNet-1K for fine-grained analysis, like ImageNet-10 that mimics the class distribution of CIFAR-10 but with high-resolution images. For hard OOD evaluation, we exploit ImageNet-20 with 20 categories similar to ImageNet-10 in the semantic space

Table 1: Comparison with competitive OOD detection baselines based on ResNet-50. The ID data are ImageNet-1K. ↑ indicates larger values are better and ↓ indicates smaller values are better.

| Method | OOD Dataset | | | | | | | | | |
| | iNaturalist | | SUN | | Places | | Textures | | Average | |
| | AUROC↑ | FPR95↓ | AUROC↑ | FPR95↓ | AUROC↑ | FPR95↓ | AUROC↑ | FPR95↓ | AUROC↑ | FPR95↓ |
|---|---|---|---|---|---|---|---|---|---|---|
| MSP | 87.74 | 54.99 | 80.86 | 70.83 | 79.76 | 73.99 | 79.61 | 68.00 | 81.99 | 66.95 |
| ODIN | 91.37 | 41.57 | 86.89 | 53.97 | 84.44 | 62.15 | 87.57 | 45.53 | 87.57 | 50.80 |
| Mahalanobis | 52.65 | 97.00 | 42.41 | 98.50 | 41.79 | 98.40 | 85.01 | 55.80 | 55.47 | 87.43 |
| Energy score | 89.95 | 55.72 | 85.89 | 59.26 | 82.86 | 64.92 | 85.99 | 53.72 | 86.17 | 58.41 |
| ReAct | 96.22 | 20.38 | 94.20 | 24.20 | 91.58 | 33.85 | 89.80 | 47.30 | 92.95 | 31.43 |
| DICE | 94.49 | 25.63 | 90.83 | 35.15 | 87.48 | 46.49 | 90.30 | 31.72 | 90.77 | 34.75 |
| DICE+ReAct | 96.24 | 18.64 | 93.94 | 25.45 | 90.67 | 36.86 | 92.74 | 28.07 | 93.40 | 27.25 |
| ASH-B | 94.25 | 28.95 | 90.32 | 40.21 | 87.52 | 49.52 | 91.53 | 33.48 | 90.91 | 39.04 |
| **ASH-B + CoVer** | 97.14 | 14.04 | 94.12 | 25.77 | 91.05 | 35.93 | 91.93 | 30.39 | 93.56 | 26.53 |
| ASH-S | 97.88 | 11.38 | 94.04 | 27.96 | 91.03 | 39.74 | **97.62** | **11.88** | 95.14 | 22.74 |
| **ASH-S + CoVer** | **98.33** | **8.73** | **94.59** | **26.63** | **91.47** | **38.06** | 97.22 | 13.92 | **95.40** | **21.83** |

Table 2: Comparison with competitive OOD detection baselines based on CLIP-B/16. The ID data are ImageNet-1K. ↑ indicates larger values are better and ↓ indicates smaller values are better.

| Method | OOD Dataset | | | | | | | | | |
| | iNaturalist | | SUN | | Places | | Textures | | Average | |
| | AUROC↑ | FPR95↓ | AUROC↑ | FPR95↓ | AUROC↑ | FPR95↓ | AUROC↑ | FPR95↓ | AUROC↑ | FPR95↓ |
|---|---|---|---|---|---|---|---|---|---|---|
| | Requires training (or w. fine-tuning) | | | | | | | | | |
| MSP | 87.44 | 58.36 | 79.73 | 73.72 | 79.67 | 74.41 | 79.69 | 71.93 | 81.63 | 69.61 |
| ODIN | 94.65 | 30.22 | 87.17 | 54.04 | 85.54 | 55.06 | 87.85 | 51.67 | 88.80 | 47.75 |
| Energy | 95.33 | 26.12 | 92.66 | 35.97 | 91.41 | 39.87 | 86.76 | 57.61 | 91.54 | 39.89 |
| GradNorm | 72.56 | 81.50 | 72.86 | 82.00 | 73.70 | 80.41 | 70.26 | 79.36 | 72.35 | 80.82 |
| ViM | 93.16 | 32.19 | 87.19 | 54.01 | 83.75 | 60.67 | 87.18 | 53.94 | 87.82 | 50.20 |
| KNN | 94.52 | 29.17 | 92.67 | 35.62 | 91.02 | 39.61 | 85.67 | 64.35 | 90.97 | 42.19 |
| | Zero-shot (no training required) | | | | | | | | | |
| Mahalanobis | 56.22 | 99.22 | 60.89 | 99.28 | 68.96 | 98.31 | 65.36 | 98.15 | 62.86 | 98.74 |
| Energy | 85.54 | 80.49 | 84.21 | 78.75 | 84.81 | 72.29 | 66.63 | 92.89 | 80.30 | 81.11 |
| ZOC | 86.09 | 87.30 | 81.20 | 81.51 | 83.39 | 73.06 | 76.46 | 98.90 | 81.79 | 85.19 |
| MCM | 94.61 | 30.95 | 92.57 | 37.57 | 89.77 | 44.65 | 86.10 | 57.77 | 90.76 | 42.73 |
| **CoVer (ours)** | **95.98** | **22.55** | **93.42** | **32.85** | **90.27** | **40.71** | **90.14** | **43.39** | **92.45** | **34.88** |

(e.g., dog (ID) vs. wolf (OOD)). To have more experimental comparison, we also reproduce one setting from spurious OOD detection [35], whose hard OOD inputs are created to share the same background (i.e., water) as ID data but have different object labels (e.g., a boat rather than a bird). To select the most effective corruption types for each method, we use SVHN [37] as the validation set.

**Model Setup.** In this paper, we implement CoVer on various architectures, including DNN-like ResNet50, and VLM-like CLIP [42], AltCLIP [6], MetaCLIP [56], GroupViT [57]. Unless otherwise instructed, for VLM-based zero-shot OOD detection, we use CLIP-B/16 which consists of an image encoder based on ViT-B/16 Transformer [15] and a text encoder built with the masked self-attention Transformer [50]. We use the algorithmically generated corruptions defined in [22]. Each type of corruption has a severity level $\epsilon$ from 1 to 5, with $\epsilon = 1$ being the least severe and increasing up to $\epsilon = 5$. By default, we use the CoVer score in the max-softmax form and set $\tau = 1$ as the temperature.

**Baseline Methods and Evaluation Metrics.** We conpare the proposed method with various competitive methods. Specifically, we adopt Maximum Softmax Probability (MSP) [23], ODIN [29], Mahalanobis [28], Energy [30], ReAct [45], DICE [46] and ASH [1] as traditional OOD detection baseline methods. The VLM-based OOD detection methods we compared with include MCM, a method specifically designed for zero-shot OOD detection, as well as some traditional methods including MSP, ODIN, Energy, Mahalanobis, GradNorm [24], ViM [51], KNN [47], ZOC [18] that were re-implemented using a finetuned CLIP ViT-B/16 on the ImageNet-1K, see Appendix A for more details. For a fair comparison, we keep the original hyperparameter setups of the comparative methods and adopt the following metrics to evaluate the OOD detection performance: (1) the false positive rate (FPR95) of the OOD samples when the true positive rate (TPR) [29] of the in-distribution samples is at 95%, (2) the area under the receiver operating characteristic curve (AUROC) [13].

Table 3: Compatibility experiments of CoVer combined with different OOD detection methods. The ID data are ImageNet-1K. ↑ indicates larger values are better and ↓ indicates smaller values are better.

| Architecture | Method | OOD Dataset | | | | | | | | | |
| | | iNaturalist | | SUN | | Places | | Textures | | Average | |
| | | AUROC↑ | FPR95↓ | AUROC↑ | FPR95↓ | AUROC↑ | FPR95↓ | AUROC↑ | FPR95↓ | AUROC↑ | FPR95↓ |
| ResNet50 | ReAct | 96.22 | 20.38 | 94.20 | 24.20 | 91.58 | 33.85 | 89.80 | 47.30 | 92.95 | 31.43 |
| | **ReAct+CoVer** | **97.58** | **13.35** | **95.7** | **18.91** | **93.08** | **29.02** | **91.55** | **40.74** | **94.48** | **25.51** |
| | DICE | 94.49 | 25.63 | 90.83 | 35.15 | 87.48 | 46.49 | 90.30 | 31.72 | 90.77 | 34.75 |
| | **DICE+CoVer** | **96.8** | **16.56** | **93.53** | **28.52** | **90.00** | **40.54** | **91.14** | **31.15** | **92.87** | **29.19** |
| | DICE (ReAct) | 96.24 | 18.64 | 93.94 | 25.45 | 90.67 | 36.86 | 92.74 | 28.07 | 93.40 | 27.25 |
| | **DICE (ReAct)+CoVer** | **97.74** | **11.38** | **94.83** | **23.44** | **91.83** | **33.87** | **93.43** | **28.95** | **94.46** | **24.41** |
| | ASH-B | 94.25 | 28.95 | 90.32 | 40.21 | 87.52 | 49.52 | 91.53 | 33.48 | 90.91 | 39.04 |
| | **ASH-B+CoVer** | **97.14** | **14.04** | **94.12** | **25.77** | **91.05** | **35.93** | **91.93** | **30.39** | **93.56** | **26.53** |
| CLIP-B/16 | MCM | 94.61 | 30.95 | 92.57 | 37.57 | 89.77 | 44.65 | 86.10 | 57.77 | 90.76 | 42.73 |
| | **MCM+CoVer** | **95.62** | **24.35** | **93.48** | **31.94** | **90.67** | **39.74** | **88.61** | **50.44** | **92.10** | **36.62** |
| | LoCoOp | 92.77 | 42.38 | 92.88 | 33.09 | 90.28 | 41.08 | 91.07 | 40.34 | 91.75 | 39.22 |
| | **LoCoOp+CoVer** | **93.07** | **41.62** | **93.71** | **31.90** | **91.03** | **38.04** | **92.90** | **32.85** | **92.68** | **36.10** |
| | CLIPN | **95.63** | **21.62** | 94.27 | 25.18 | 93.15 | 30.51 | **90.34** | 41.68 | 93.35 | 29.66 |
| | **CLIPN+CoVer** | 95.41 | 23.14 | **95.72** | **17.13** | **94.80** | **23.05** | 88.59 | 40.82 | **93.63** | **26.04** |
| | NegLabel | 99.49 | 1.93 | **95.46** | **20.95** | 91.58 | 36.45 | 89.89 | 45.12 | 94.10 | 26.11 |
| | **NegLabel+CoVer** | **99.59** | **1.15** | 94.56 | 28.84 | **95.01** | **25.65** | **92.39** | **40.39** | **95.39** | **24.01** |

## 4.2 Main Results

**Overall results of OOD detection performance comparison with different baselines.** To evaluate the effectiveness of CoVer, we compare it with existing baseline OOD detection methods on the ImageNet-1K benchmark in two aspects. In Table 1, we present the performance comparison with traditional OOD detection methods using ResNet-50 as the backbone. Our CoVer combined with ASH-S can achieve better OOD detection performance, which verifies the effectiveness of our method with an average multi-dimensional estimated confidence score. In Table 2, we provide the results compared with VLM-based OOD detection methods, which are classified into two groups: fine-tuning methods that require extra data for fine-tuning CLIP and zero-shot methods that require no training. Our method CoVer consistently achieves better performance across the four OOD datasets, aligning with the analysis that OOD and ID data are better distinguished under the expanded dimensions.

**Compatibility experiments of CoVer combined with different OOD detection methods.** Since it is a simple design for representation dimension expansion, CoVer can be easily integrated into previous OOD detection methods and achieve performance improvements. In Table 3, we first consider some methods with minor modifications to the architecture of ResNet50, e.g., ReAct, DICE and ASH. While fixing the detection models, we replace each of their OOD scores with our CoVer score. Then we integrate the proposed CoVer into several VLM-based OOD detection methods. For LoCoOp [36], a CLIP-fine-tuning method with the MCM score, we follow its fine-tuning strategy and report the results with the CoVer's scoring mode. For CLIPN [52] and NegLabel [26] that both introduce a new OOD score, we redesigned them based on the critical idea of confidence average, as detailed in the Appendix C.3. Our CoVer can consistently help these methods gain better performance without specific modality limitations, which shows the algorithmic robustness of our proposed method. More results with other methods and further discussions and analyses are provided in Appendix C.2.

**Zero-shot OOD detection performance comparison on hard OOD detection.** Following the settings in [31, 26], we explore the superiority of our proposed CoVer compared with MCM [31] on hard OOD detection tasks, as shown in Table 4. Specifically, we alternately use ImageNet-10 and ImageNet-20, ImageNet-10 and ImageNet-100 as ID and OOD data to simulate the setups presented in [19]. The results demonstrate CoVer has a better distinguished ability than MCM for semantically hard OOD data. We also report the experimental data on zero-shot spurious OOD detection task in the last row of Table 4, which also shows the better performance of the proposed CoVer than MCM.

## 4.3 Ablation and Future Discussions

In this part, we conduct extensive ablation studies and provide a thorough understanding of our CoVer. The extra results and discussions (e.g., impact and limitations) are provided in Appendix C.2.

**Ablation on diverse VLM architectures.** We compare our CoVer score with the MCM score on different VLM architectures and the results are reported in Table 5. The first part shows the superiority

Table 4: Zero-shot OOD detection performance comparison on hard OOD detection tasks. All methods are based on CLIP-B/16.

| ID Dataset | OOD Dataset | Method | AUROC↑ | FPR95↓ |
|---|---|---|---|---|
| ImageNet-10 | ImageNet-20 | MCM | 98.60 | 6.00 |
| | | CoVer | **98.68** | **4.10** |
| ImageNet-20 | ImageNet-10 | MCM | **97.69** | 17.07 |
| | | CoVer | 97.60 | **14.58** |
| ImageNet-10 | ImageNet-100 | MCM | **99.30** | 2.30 |
| | | CoVer | 99.28 | **1.92** |
| ImageNet-100 | ImageNet-10 | MCM | **86.50** | 66.18 |
| | | CoVer | 86.38 | **65.55** |
| WaterBirds | Spurious OOD | MCM | 90.31 | 35.66 |
| | | CoVer | **90.52** | **33.17** |

Table 5: Zero-shot OOD detection performance with different VLM architectures' representations on ImageNet-1K(ID).

| Architecture | Backbone | Method | AUROC↑ | FPR95↓ |
|---|---|---|---|---|
| CLIP | ResNet50 | MCM | 88.99 | 49.79 |
| | | CoVer | **89.98** | **46.18** |
| | ViT-B/32 | MCM | 89.82 | 45.75 |
| | | CoVer | **90.21** | **44.78** |
| | ViT-L/14 | MCM | 91.49 | 38.17 |
| | | CoVer | **92.61** | **32.97** |
| AltCLIP | ViT-L/14 | MCM | 91.54 | 40.74 |
| | | CoVer | **93.03** | **32.15** |
| MetaCLIP | VIT-B/16-quickgelu | MCM | 87.57 | 58.97 |
| | | CoVer | **88.64** | **55.68** |
| GroupViT | GroupViT | MCM | 85.10 | 57.85 |
| | | CoVer | **86.94** | **51.19** |

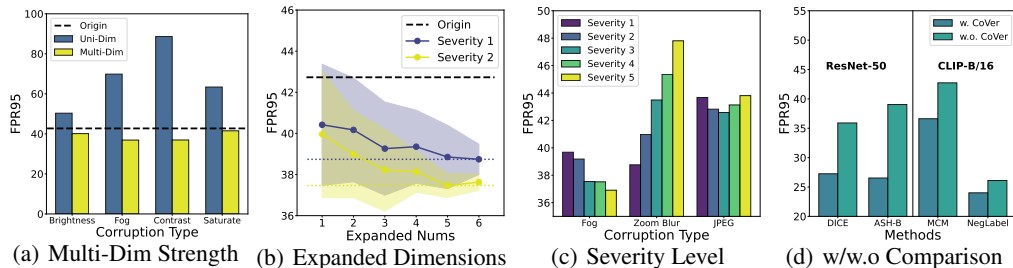

(a) Multi-Dim Strength    (b) Expanded Dimensions    (c) Severity Level    (d) w/w.o Comparison

Figure 5: Ablation Study. (a) superiority of the multi-dimensional scoring framework; (b) exploration of different quantity of expanded input dimensions; (c) using different severity levels of a specific corruption type; (d) comparison with different realizations for each dimensional confidence score.

of our CoVer on different backbones of CLIP, and the larger backbone boosts the performance of OOD detection. The second part shows that CoVer can generalize better to various VLM architectures compared to MCM. More fine-grained results on different OOD sets can be found in Appendix C.2.1.

**Superiority of multi-dimensional scoring framework.** To verify the superiority of our CoVer with a multi-dimensional scoring framework compared with the uni-dimensional one, we report the performance comparison using different corruption types in Figure 5(a). Here, the uni-dimensional framework is denoted as using a single corrupted input for confidence estimation. The results indicate that extended dimensions provide valuable clues for enhancing OOD discriminative representations, thus enlarging the separability between ID and OOD data. We leave the fine-grained results and further discussions in Appendix C.2.2, which provide a better understanding of the improvement.

**Significance of the number of expanded measuring dimensions.** As a critical aspect of our CoVer, the number of the extended confidence measuring dimensions will control the performance enhancement for OOD detection. In Figure 5(b), we present results for varying the number of expanded dimensions using various corruptions with the same severity level. The results demonstrate that an increasing number of expanded representation dimensions gradually improves the performance then probably declines, while consistently outperforming the baseline. This indicates that the addition of measuring dimensions prioritizes enhancing the distinction of OOD data with mutated confidence while ID data shows resistance. More detailed results and analyses are provided in Appendix C.2.3.

**Comparison of the variation trends in different corruption severity levels.** In Figure 5(c), we show the performance by varying the severity level $\epsilon$ for each specific corruption style. We can observe that three types of trends emerge with an increasing level of severity, i.e., up, down, and up then down. This phenomenon indicates that an appropriate level of corruption is critical for the optimization of CoVer. One possible reason may be that the threshold maximizing the distinction between ID and OOD data varies from different types of corruption. To further explain this observation, we provide more results on the $\epsilon$ among various corruption styles and more discussions in Appendix C.2.4.

**Generality of integrating with different OOD detection schemes.** Since the proposed CoVer is a general scoring framework with an average of confidence scores measured from multiple dimensions, the specific realization for each dimensional score have multiple choices. In Figure 5(d), we present the comparison with different realizations integrated w./w.o. CoVer (e.g., ResNet-50 based DICE [46] and ASH-B [1], CLIP-B/16 based MCM [31] and NegLabel [26]), where they have different performance improvements compared with the original baseline without constraints for the modality.

## 5 Discussion

### 5.1 Broader Impact

OOD detection is crucial for deploying reliable deep learning systems in real-world applications [21], ensuring models can effectively identify data that differ significantly from the training distribution. This ability is vital in safety-critical areas [5], where incorrect predictions due to unexpected inputs can lead to severe consequences. For instance, in the field of autonomous driving, OOD detection helps the system recognize and react appropriately to novel scenarios not covered during training, such as new road signs or altered traffic conditions due to construction. This is imperative as it prevents autonomous vehicles from making potentially hazardous decisions based on learned but now irrelevant data, thereby enhancing their safety and robustness in dynamic environments.

Our research highlights a fundamental yet overlooked challenge in existing OOD detection methods, which often specialize in a single input type. This specialization may inadvertently limit the representational dimensions for detection, complicating the identification of subtle OOD samples. For effective OOD detection, it is crucial to not only improve empirical performance through enhanced OOD discriminative representations but also to address this pervasive issue within the general scoring framework. Our new scoring framework leverages expanded input dimensions and utilizes a confidence score expectation to address these concerns, which also shares similar intuitions with some related work [43] in adversarial defense via random transformation. Comprehensive experiments demonstrate its effectiveness and compatibility, suggesting that our method is potentially a new generalized framework and provides new insights into OOD detection from a different perspective.

### 5.2 Limitation

While our method introduces a promising framework for OOD detection and provides unique insights through the use of corrupted images to enhance representational dimensions, it does face certain challenges. First, our method indeed faces several failure cases. When CoVer utilizes certain severe corruption types (e.g., Spatter, Elastic transform), its performance is worse than with single input. This is because these types are more severe compared to others, leading to excessive damages to semantic features. Effective corruption types are those only perturb non-semantic features, which generally exist at the high-frequency level, resulting in different confidence variations between ID and OOD data. Notably, except for leveraging the validation set, our approach lacks a standardized criterion for selecting the types and intensities of corruptions, which is essential for uniform effectiveness across various scenarios. Additionally, the expansion of input dimensions, though beneficial for detection accuracy, may lead to increased evaluation times. These limitations highlight areas for potential improvement, particularly in balancing detection capabilities with computational efficiency. Despite these challenges, the enhanced detection capabilities our method introduces mark a significant step forward in the reliability of machine learning models against OOD inputs.

## 6 Conclusion

In this paper, we introduce a novel perspective for OOD detection, i.e., expanding the representation dimensions. With the different common corruptions, we reveal an interesting phenomenon termed *confidence mutation*, where the confidence values of some overconfident OOD samples can vary significantly compared with the original inputs. To this end, we propose a new scoring framework, namely, *Confidence aVerage* (CoVer), which simultaneously considers the original and expanded input dimensions. Adopting a simple but effective average operation, CoVer can capture the dynamical discrimination of OOD samples and better enhance the separability of ID and OOD distributions. We have conducted extensive experiments to present its effectiveness and compatibility with different methods. We hope our work can draw new insights from a different view on OOD detection.

# 7 Acknowledgement

BXZ, ZMW and BD were supported by the National Key Research and Development Program of China 2023YFC2705700, National Natural Science Foundation of China under Grants 62225113, 62271357, Natural Science Foundation of Hubei Province under Grants 2023BAB072, the Innovative Research Group Project of Hubei Province under Grants 2024AFA017, the Fundamental Research Funds for the Central Universities under Grants 2042023kf0134, Wuhan Natural Science Foundation 2024040801020236, and the numerical calculations in this paper have been done on the supercomputing system in the Supercomputing Center of Wuhan University. JNZ and BH were supported by NSFC General Program No. 62376235, Guangdong Basic and Applied Basic Research Foundation Nos. 2022A1515011652 and 2024A1515012399, RIKEN Collaborative Research Fund, HKBU Faculty Niche Research Areas No. RC-FNRA-IG/22-23/SCI/04, and HKBU CSD Departmental Incentive Scheme.

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

# Appendix for CoVer

The whole Appendix is organized as follows. In Appendix A, we present the detailed definitions and implementation of baseline methods that are considered in our exploration. In Appendix B, we provide detailed discussions about related works. In Appendix C, we provide extra experimental details and more comprehensive results with further discussion on the underlying implications. In Appendix D, we provide some preliminary statistical analysis about CoVer. Finally, in Appendix E, we provide the further analysis for a better understanding of our work.

# Reproducibility Statement

To ensure the reproducibility of experimental results, we provide the source code at `https://github.com/tmlr-group/CoVer`. Below we summarize several important aspects to facilitate reproducible results:

- **Datasets.** The datasets we used are all publicly accessible, which is introduced in Section 4. Following MCM [31], we also use subsets of ImageNet-1K for fine-grained analysis, like ImageNet-10. For hard OOD evaluation, we exploit ImageNet-20 with 20 categories similar to ImageNet-10. We also reproduce the spurious OOD detection [35] with $r = 0.9$, which determines relative size of majority vs. minority groups.

- **Assumption.** We set our main experiments to a zero-shot scenario where a well-trained CLIP-like model on the original classification task is available [42]. Under this assumption, CoVer detects OOD samples in parallel with the zero-shot classification task and has no impact on ID classification performance.

- **Open source.** The source code is available at `https://github.com/tmlr-group/CoVer`. We provide a backbone for our experiments as well as several auxiliary components, such as score estimation.

- **Environment.** All experiments are conducted on NVIDIA GeForce RTX 3090 GPUs with Python 3.10 and PyTorch 2.2.

# A Details about Considered Baselines and Metrics

In this section, we provide details about the baselines for the scoring functions, as well as the corresponding hyper-parameters and other related metrics that are considered in our work.

**Maximum Softmax Probability (MSP).** [23] proposes to use maximum softmax probability to discriminate ID and OOD samples. The score is defined as follows,

$$S_{\text{MSP}}(x; f) = \max_c P(y = c|x; f) = \max \texttt{softmax}(f(x)), \tag{7}$$

where $f$ represents the given well-trained model and $c$ is one of the classes $\mathcal{Y} = \{1, \ldots, C\}$. The larger softmax score indicates the larger probability for a sample to be ID data, reflecting the model's confidence on the sample.

**ODIN.** [29] designed the ODIN score, leveraging the temperature scaling and tiny perturbations to widen the gap between the distributions of ID and OOD samples. The ODIN score is defined as follows,

$$S_{\text{ODIN}}(x; f) = \max_c P(y = c|\tilde{x}; f) = \max \texttt{softmax}(\frac{f(\tilde{x})}{T}), \tag{8}$$

where $\tilde{x}$ represents the perturbed samples (controled by $\epsilon$), $T$ represents the temperature. For fair comparison, we adopt the suggested hyperparameters [29]: $\epsilon = 1.4 \times 10^{-3}$, $T = 1.0 \times 10^4$.

**Mahalanobis.** [28] introduces a Mahalanobis distance-based confidence score, exploiting the feature space of the neural networks by inspecting the class conditional Gaussian distributions. The Mahalanobis distance score is defined as follows,

$$S_{\text{Mahalanobis}}(x; f) = \max_c -(f(x) - \hat{\mu}_c)^T \hat{\Sigma}^{-1}(f(x) - \hat{\mu}_c), \tag{9}$$

where $\hat{\mu}_c$ represents the estimated mean of multivariate Gaussian distribution of class $c$, $\hat{\Sigma}$ represents the estimated tied covariance of the $C$ class-conditional Gaussian distributions.

**Energy.** [30] proposes to use the Energy of the predicted logits to distinguish the ID and OOD samples. The Energy score is defined as follows,

$$S_{\text{Energy}}(x; f) = -T \log \sum_{c=1}^{C} e^{f(x)_c/T}, \tag{10}$$

where $T$ represents the temperature parameter. As theoretically illustrated in [30], a lower Energy score indicates a higher probability for a sample to be ID. Following [30], we fix the $T$ to $1.0$ throughout all experiments.

**ASH.** [1] designs a extremely simple, post-hoc method called Activation SHaping for OOD detection. It removes a large portion of an input's activation at a late layer and adjusts the rest of the activation values by scaling them up or assigning them a constant value. The simplified representation is then passed throughout the rest of the network. The logit output is used to classify ID samples and calculate scores for OOD detection as usual. For ASH-B version, we adopt the MSP score and implement it with the hyperparameter $p = 65$; For ASH-S version, we apply it with energy score and the hyperparameter $p = 90$. Both settings are suggested by [1].

**MCM.** [31] proposes Maximum Concept Matching (MCM), a simple yet effective zero-shot OOD detection method based on aligning visual features with textual concepts. Formally, the MCM score can be defined as:

$$S_{\text{MCM}}(\mathbf{x}'; \mathcal{Y}_{\text{in}}, \mathcal{T}, \mathcal{I}) = \max_{i} \frac{e^{s_i(\mathbf{x}')/\tau}}{\sum_{j=1}^{K} e^{s_j(\mathbf{x}')/\tau}} \tag{11}$$

**CLIPN.** [52] proposes a novel CLIP architecture, which equips CLIP with a "no" logic via the learnable "no" prompts and a "no" text encoder. Specifically, CLIPN proposes two novel inference algorithms to perform OOD detection via using negation semantics, where the algorithm named agreeing-to-differ (ATD) is more effective in experimental results. The ATD form of the CLIPN score can be formulated as follows,

$$S_{\text{CLIPN}}(x) = \sum_{j=1}^{C} \frac{e^{s_{i,j}^{no}(x)/\tau}}{e^{s_{i,j}(x)/\tau} + e^{s_{i,j}^{no}(x)/\tau}} \cdot \frac{e^{s_{i,j}(x)/\tau}}{\sum_{k=1}^{C} e^{s_{i,k}(x)/\tau}} \tag{12}$$

where C is the number of classes, $s_{i,j}(x)$ and $s_{i,j}^{no}(x)$ are denoted as the inner product of the image feature and the corresponding text feature

$$s_{i,j}(x) = <\mathbf{f}^{\text{image}}(x), \mathbf{f}^{\text{text}}(p(y_j))>, \quad s_{i,j}^{no}(x) = <\mathbf{f}^{\text{image}}(x), \mathbf{f}_{\text{no}}^{\text{text}}(p(y_j^{no}))> \tag{13}$$

where $\mathbf{f}_{\text{no}}^{\text{text}}$ is the "no" text encoder and $p(y_j^{no})$ the text with "no" logic.

**NegLabel.** [26] proposes a novel post hoc OOD detection method, called NegLabel, which takes a vast number of negative labels from extensive corpus databases and designs a novel scheme for the OOD score collaborated with negative labels. NegLabel score can be formulated as

$$S_{\text{NegLabel}}(\mathbf{x}) = S^* \left( \text{sim}(\mathbf{x}, \mathcal{Y}), \text{sim}(\mathbf{x}, \mathcal{Y}^-) \right) \tag{14}$$

where $S^* (\cdot, \cdot)$ represents a fusion function that combines the similarity of a sample with ID labels $\text{sim}(\mathbf{x}, \mathcal{Y})$ and the similarity of the sample with negative labels $\text{sim}(\mathbf{x}, \mathcal{Y}^-)$. The sum-softmax form of NegLabel score is defined as follows,

$$S_{\text{NegLabel}}(x) = \frac{\sum_{i=1}^{K} e^{s_i(x)/\tau}}{\sum_{i=1}^{K} e^{s_i(x)/\tau} + \sum_{j=1}^{M} e^{s_j^{\text{neg}}(x)/\tau}} \tag{15}$$

where $K$ is the number of ID labels, $\tau$ is the temperature coefficient of the softmax function and $M$ is the number of negative labels, $s_i(x)$ and $s_j^{\text{neg}}(x)$ are formulated as the cosine similarity, defined as follows,

$$s_i(x) = \cos(\mathbf{f}^{\text{image}}(x), \boldsymbol{e}_i), \quad s_j^{\text{neg}}(x) = \cos(\mathbf{f}^{\text{image}}(x), \widetilde{e}_j) \tag{16}$$

# B    Detailed Discussion with Related Works

In this section, we provide detailed discussions about related works.

**Traditional OOD Detection.**    There has been an increasing interest in OOD detection since the phenomenon of overconfidence in OOD data was first discovered in [38]. As a formal benchmark for OOD detection, [23] proposed using softmax prediction probability as a conventional baseline method. Afterward, numerous approaches have been developed to address visual OOD detection, which can be classified into two categories, i.e., post hoc scoring mechanism and training-time regularization [59, 58]. Post hoc methods are dedicated to exploring a better OOD score by freezing the model's parameters. ODIN [29] improves the previous MSP [23] by scaling with the temperature and slightly perturbing the inputs. Mahalanobis introduces the Mahalanobis distance in the feature space to measure the confidence score. Energy [30] exploits the energy function [27] to distinguish ID and OOD data. Both ReAct [45] and DICE [46] are improved from Energy, ReAct by feature clipping, and DICE by discarding the most prominent weights in the fully connected layer. ASH [1] designs an extremely simple method that removes a large portion of an input's activation and adjusts the rest. On the other hand, training-time regularization methods exploit the potential access to partial OOD data during model training. MOS [24] groups all classes and introduces an extra class to each group to reformulate the loss function during training. VOS [17] enhances the quality of the energy score by creating synthetic virtual anomalies. CIDER [34] exploits KNN [47] to boost OOD detection performance through the optimization of contrastive loss. DAOL [53] alleviates the OOD distribution discrepancy by crafting an OOD distribution set that contains all distributions in a Wasserstein ball centered on the auxiliary OOD distribution. The presence of outliers leads to superior performance compared to training without outliers, as evidenced by numerous previous studies [3, 25, 2, 16, 62].

**OOD Detection with vision-language representations.**    With the rapid development of multi-modal large language models (MLLMs), such as CLIP [42], much attention has been paid to OOD detection with vision-language representations [33]. MCM [31] proposed the first zero-shot OOD detection framework that combines the temperature scaling strategy and maximum softmax probability as the OOD score. Following MCM, some works fine-tuned CLIP's image encoder for visual OOD detection [48, 36]. NPOS [48] utilized generated OOD data to optimize the ID-OOD decision boundary. LoCoOp exploited the portions of CLIP's local features as OOD features to realize OOD regularization. Some latest methods [52, 26] boosted OOD detection by adding extra clues obtained from negative textual information. CLIPN [52] equipped CLIP with a "no" logic via a text encoder that can understand negative prompts. NegLabel [26] introduced numerous negative labels and distinguished OOD samples by examining their affinities between ID and negative labels.

**Data depths and information projections.**    Computing OOD scores on the embedding output of the last layer of the encoder is not the best choice for textual OOD detection. To address this, [12] proposed aggregating OOD scores across all layers and introduced an extended text OOD classification benchmark, MILTOOD-C. In a similar vein, RainProof [11] introduced a relative information projection framework and a new benchmark called LOFTER on text generators, considering both OOD performance and task-specific metrics. Building on the idea of information projection, REFEREE [40] leveraged I-projection to extract relevant information from the softmax outputs of a network for black-box adversarial attack detection. On the other hand, APPROVED [41] proposed to compute a similarity score between an input sample and the training distribution using the statistical notion of data depth at the logit layer. HAMPER [9] introduced a method to detect adversarial examples by utilizing the concept of data depths, particularly the halfspace-mass (HM) depth, known for its attractive properties and non-differentiability. Furthermore, TRUSTED [8] relied on the information available across all hidden layers of a network, leveraging a novel similarity score based on the Integrate Rank-Weighted depth for textual OOD detection. LAROUSSE [10] employed a new anomaly score built on the HM depth to detect textual adversarial attacks in an unsupervised manner.

# C    Additional Experimental Results and Further Discussion

In this section, we provide additional experimental results from different perspectives to verify the effectiveness our proposed CoVer. First, we introduce the additional experimental setups for the empirical verification in previous figures and implement details about our method. Second, we

provide comprehensive results with further discussions of ablations. Finally, extensive visualization analyses are provided for a better understanding of CoVer.

## C.1 Additional Experiment Setups

**Implement Details.** Unless otherwise specified (e.g. Table 1 and Table 3), we conduct the major experiments based on pre-trained CLIP-B/16 for zero-shot OOD detection, following the previous research work [31, 26]. Furthermore, the primary form of CoVer is based on the maximum-softmax score function, as defined in Section 3.3. For the extended corrupted inputs, We utilize the SVHN dataset as the validation set to determine the most effective corruption types for each method in all experiments. The detailed adopted corruption types for each method are provided in Table 6, where *Corruption Type*(X) denote the corruption at X severity level.

Table 6: Adopted corruption types and corresponding severity levels when CoVer integrated with other methods.

| Method | Expanded Corruption Types |
|---|---|
| ReAct + CoVer | *Contrast*(3) |
| DICE + CoVer | *Brightness*(1, 2), *Gaussian Blur*(1, 2), *Saturate*(1, 2), and *Fog*(1, 2) |
| DICE (ReAct) + CoVer | *Brightness*(1, 2) |
| ASH-B / ASH-S + CoVer | *Brightness*(1, 2) |
| MCM + CoVer | *Brightness*(1, 2), *Gaussian Blur*(1, 2), *Motion Blur*(1, 2), *Saturate*(1, 2), *Defocus Blur*(1, 2), and *Fog*(1, 2) |
| LoCoOp + CoVer | *Brightness*(1, 2), *Gaussian Blur*(1, 2), *Motion Blur*(1, 2), *Saturate*(1, 2), *Defocus Blur*(1, 2), and *Fog*(1, 2) |
| CLIPN + CoVer | *Brightness*(1) and *Saturate*(1) |
| NegLabel + CoVer | *Brightness*(1, 2) and *Saturate*(1, 2) |

**Figure 1.** In Figure 1, we compare the score distributions and detection results with different input modes. Specifically, we use the corruption type of *Contrast*(4) as an example and report the results on the *iNaturalist* dataset of the ImageNet-1K benchmark. In the right panel of 1, we realize our CoVer by averaging the confidence scores obtained by the original and corrupted(*Contrast*(4)) inputs.

**Figure 2.** In Figure 2, we conduct experiments to demonstrate the detailed explanations for the discovery illustrated in Figure 1. For the fair comparison, all samples here, including confident ID, unconfident ID, overconfident OOD, and unconfident OOD samples, are randomly sampled from the ID and OOD distributions in the left and middle panels of Figure 1.

**Figure 3.** In Figure 3, we visualize several samples for further understanding of the confidence mutation. Specifically, we use the Fast Fourier transform (FFT) to obtain the low-frequency and high-frequency portions of the image with the radius of circular filter $r = 0.6$. Same as Figure 1 and Figure 2, unconfident ID data and overconfident OOD data are randomly sampled from the corresponding part of ImageNet and iNaturalist datasets, respectively. We continue to adopt *Contrast*(4) for corruption.

**Figure 5.** For Figure 5(a) to Figure 5(d), we provide discussions about the detailed experimental settings and their fine-grained results in Appendix C.2.

## C.2 Full Results of Ablations

### C.2.1 Ablation on VLM Architectures.

The detailed results are shown in Table 7. It is evidenced that in addition to CLIP Vit-B/16, our CoVer can boost the OOD detection performance on various kinds of VLM architectures compared to MCM, including the CLIP architecture based on different backbone networks (e.g., ResNet50 and Vit-L/14) and other types of VLM architectures (e.g., AltCLIP, MetaCLIP, and GroupViT).

### C.2.2 Superiority of Multi-Dimensional Scoring Framework.

In Figure 5(a), we present some representative results to verify the superiority of the multi-dimensional scoring framework. Here, we report the fine-grained results using different types of corruptions with a severity level of 5, as shown in Table 8. We can find that there is constantly a bad OOD detection

Table 7: Compared to MCM with different VLM architectures on ImageNet-1K(ID). All values are percentages. ↑ indicates larger values are better and ↓ indicates smaller values are better.

| Architecture | Backbone | Method | iNaturalist | | SUN | | Places | | Textures | | Average | |
|---|---|---|---|---|---|---|---|---|---|---|---|---|
| | | | AUROC↑ | FPR95↓ | AUROC↑ | FPR95↓ | AUROC↑ | FPR95↓ | AUROC↑ | FPR95↓ | AUROC↑ | FPR95↓ |
| CLIP | ResNet50 | MCM | 93.86 | 32.16 | 90.74 | 46.21 | 85.66 | 60.68 | 85.71 | 60.11 | 88.99 | 49.79 |
| | | **CoVer** | **94.57** | **30.26** | **90.75** | **45.51** | **86.92** | **54.84** | **87.66** | **54.11** | **89.97** | **46.18** |
| | ViT-B/32 | MCM | **93.61** | **33.92** | 91.42 | 41.79 | 89.56 | 45.64 | 84.67 | 61.63 | 89.82 | 45.75 |
| | | **CoVer** | 93.52 | 34.51 | **91.66** | **40.65** | **88.89** | **46.92** | **86.77** | **57.06** | **90.21** | **44.78** |
| | ViT-B/16 | MCM | 94.61 | 30.95 | 92.57 | 37.57 | 89.77 | 44.65 | 86.10 | 57.77 | 90.76 | 42.73 |
| | | **CoVer** | **95.62** | **24.35** | **93.48** | **31.94** | **90.67** | **39.74** | **88.61** | **50.44** | **92.10** | **36.62** |
| | ViT-L/14 | MCM | 94.95 | 28.38 | 94.14 | 29.0 | 92.0 | 35.42 | 84.88 | 59.88 | 91.49 | 38.17 |
| | | **CoVer** | **96.16** | **20.84** | **94.91** | **24.58** | **92.37** | **32.4** | **87.0** | **54.04** | **92.61** | **32.96** |
| AltCLIP | ViT-L/14 | MCM | 92.91 | 43.31 | 94.56 | 28.5 | 91.65 | 37.92 | 87.06 | 53.24 | 91.54 | 40.74 |
| | | **CoVer** | **96.00** | **22.21** | **95.17** | **24.35** | **92.04** | **34.61** | **88.93** | **47.43** | **93.04** | **32.15** |
| MetaCLIP | ViT-B/16-quickgelu | MCM | 87.68 | 64.97 | 90.57 | 48.79 | 86.63 | 59.46 | 85.40 | 62.64 | 87.57 | 58.97 |
| | | **CoVer** | **89.30** | **61.67** | **91.09** | **46.76** | **88.07** | **54.13** | **86.09** | **60.16** | **88.64** | **55.68** |
| GroupViT | ViT-L/14 | MCM | 89.58 | 49.08 | 85.78 | 58.57 | 82.01 | 64.84 | 83.01 | 58.92 | 85.09 | 57.85 |
| | | **CoVer** | **91.65** | **37.08** | **87.67** | **53.1** | **83.74** | **60.43** | **84.7** | **54.15** | **86.94** | **51.19** |

performance under the single-dimensional scoring framework with a single corrupted image as the input sample. However, considering the multiple inputs, i.e., both original and corrupted inputs, averaging their confidence scores shows a huge performance enhancement. For example, comparing the uni-dimensional and multi-dimensional results when the corruption type is *Contrast*, the incorporation of the extended dimension provides +17.57% AUROC (from 74.20% to **91.77%**) and -52.02% FPR95 (from 88.61% to **36.95%**), indicating that corrupted inputs require the combination of original inputs to reveal its ability to enhance the distinguishability of OOD samples at the feature level. For more detailed experiments regarding CoVer expanded by corrupted inputs at other severity levels, please refer to Appendix C.2.4.

To have a more intuitive understanding on the superiority of multi-dimensional inputs, we provide a comparison with score distributions and detection results w/w.o. an expanded input dimension transformed by *Brightness*, *Fog*, *Motion blur*, and *Speckle noise* in Figure 6. From the middle column, we notice that the OOD samples are more difficult to detect by the model confidence from single corrupted inputs. This is mainly because the confidence of the ID sample, which was originally high, drops drastically when it is corrupted thereby interfering with the model's judgment of the OOD sample. In contrast, through a simple but critical average operation, CoVer generally achieves better ID-OOD separability. This phenomenon can be attributed to two main reasons. First, the ID samples have an overall higher confidence expectation, eliminating the originally confident ID interference present under the single corrupted input. Secondly, as illustrated in Figure 2 and Figure 3, the data lies in the originally overlapped part of the ID and OOD distributions, i.e., unconfident ID data and overconfident OOD data, demonstrate significant differences in the variation of confidences under the same corruption. Specifically, the ID data shows resistance while the OOD data shows vulnerability, thus better exposing the OOD samples to be rejected.

### C.2.3   Imapact of the Number of Expanded Measuring Dimensions.

In Figure 5(b), we report the OOD detection performance variations evaluated on different numbers of expanded measuring dimensions. However, we have not specifically analyzed the impact of employing different types of corruptions under each number of extended representation dimensions. In Table 9, we further present the detailed results for different numbers of extended dimensions under corruption severity levels 1 and 2, with each number we enumerate two different kinds of combinations of corruption types. The experimental results demonstrate that considering confidence estimation on both original input and expanded variant dimension constantly enhances the OOD detection performance across four OOD datasets.

However, as the expanded dimension gives priority to the phenomenon of confidence mutation that is more discriminative in OOD data than that is in ID data, the newly added representation dimension sometimes leads to a slight decline in performance. For instance, the best performance is three expanded dimensions that grouped by *Defocus blur*, *Motion blur* and *Fog* inputs under the corruption severity level 1; the combination of five types of corruptions including *Defocus blur*, *Motion blur*, *Gaussion blur*, *Fog*, and *Saturate* achieve a better OOD detection performance when the variants are generated by corruptions at severity level 2. This phenomenon suggests that, while

Table 8: Fine-grained results of CoVer using different types of corruptions with a severity level of 5 based on CLIP-B/16. The experiments are conducted on ImageNet-1k benchmark

| Corruption Type | Mode | iNaturalist AUROC↑ | iNaturalist FPR95↓ | SUN AUROC↑ | SUN FPR95↓ | Places AUROC↑ | Places FPR95↓ | Textures AUROC↑ | Textures FPR95↓ | Average AUROC↑ | Average FPR95↓ |
|---|---|---|---|---|---|---|---|---|---|---|---|
| Brightness | Uni | 92.84 | 41.42 | 90.17 | 50.25 | 87.35 | 54.42 | 86.48 | 55.21 | 89.21 | 50.33 |
| | Multi | 95.11 | 28.29 | 92.76 | 36.84 | 90.05 | 43.08 | 88.08 | 52.3 | 91.5 | 40.13 |
| Fog | Uni | 88.42 | 60.34 | 84.71 | 75.11 | 81.34 | 78.46 | 82.0 | 65.48 | 84.12 | 69.85 |
| | Multi | 95.61 | 24.19 | 93.19 | 33.77 | 90.15 | 40.96 | 88.11 | 48.72 | 91.77 | 36.91 |
| Contrast | Uni | 82.0 | 87.94 | 71.36 | 89.42 | 67.68 | 91.01 | 75.77 | 86.06 | 74.2 | 88.61 |
| | Multi | 96.41 | 19.6 | 92.46 | 36.05 | 89.17 | 44.07 | 89.05 | 48.07 | 91.77 | 36.95 |
| Motion Blur | Uni | 82.12 | 71.66 | 72.48 | 90.65 | 67.88 | 92.11 | 76.12 | 73.12 | 74.65 | 81.89 |
| | Multi | 94.98 | 26.13 | 91.4 | 42.02 | 87.78 | 50.18 | 87.24 | 50.92 | 90.35 | 42.31 |
| Defocus Blur | Uni | 80.75 | 65.67 | 77.1 | 80.25 | 74.12 | 83.31 | 79.03 | 70.0 | 77.75 | 74.81 |
| | Multi | 94.04 | 30.8 | 92.12 | 39.58 | 89.09 | 45.25 | 88.32 | 49.8 | 90.89 | 41.36 |
| Gaussian Blur | Uni | 77.49 | 73.47 | 73.84 | 83.98 | 71.13 | 86.05 | 77.61 | 69.86 | 75.02 | 78.34 |
| | Multi | 93.52 | 33.3 | 91.59 | 42.03 | 88.64 | 47.39 | 88.01 | 50.25 | 90.44 | 43.24 |
| Spatter | Uni | 75.45 | 87.44 | 81.57 | 83.09 | 79.15 | 83.43 | 71.27 | 85.57 | 76.86 | 84.88 |
| | Multi | 92.06 | 44.98 | 92.3 | 40.42 | 89.46 | 46.1 | 83.89 | 62.36 | 89.43 | 48.47 |
| Saturate | Uni | 88.48 | 57.4 | 86.37 | 64.72 | 84.76 | 66.84 | 83.03 | 64.63 | 85.66 | 63.4 |
| | Multi | 94.24 | 31.27 | 92.11 | 38.41 | 89.9 | 43.08 | 87.23 | 53.37 | 90.87 | 41.53 |
| Elastic Transform | Uni | 54.9 | 98.67 | 65.13 | 95.6 | 65.08 | 94.5 | 47.27 | 97.09 | 58.1 | 96.47 |
| | Multi | 90.57 | 53.15 | 91.12 | 45.97 | 88.52 | 49.36 | 77.98 | 72.84 | 87.05 | 55.33 |
| JPEG Compression | Uni | 79.43 | 86.23 | 83.88 | 75.1 | 80.99 | 75.62 | 77.05 | 81.49 | 80.34 | 79.61 |
| | Multi | 93.08 | 39.52 | 92.95 | 36.21 | 89.97 | 42.17 | 86.27 | 57.34 | 90.57 | 43.81 |
| Pixelate | Uni | 82.88 | 74.39 | 78.18 | 85.36 | 75.9 | 88.07 | 73.27 | 83.65 | 77.56 | 82.87 |
| | Multi | 94.56 | 29.86 | 91.79 | 39.89 | 89.11 | 45.89 | 85.86 | 58.48 | 90.33 | 43.53 |
| Speckle Noise | Uni | 83.54 | 80.59 | 69.26 | 95.65 | 67.8 | 94.86 | 68.17 | 92.07 | 72.19 | 90.79 |
| | Multi | 96.49 | 19.3 | 91.51 | 44.35 | 88.92 | 49.75 | 85.44 | 56.95 | 90.59 | 42.59 |
| Glass Blur | Uni | 76.1 | 82.53 | 74.05 | 84.98 | 70.49 | 87.97 | 62.0 | 86.05 | 70.66 | 85.38 |
| | Multi | 94.51 | 30.81 | 92.23 | 37.47 | 89.12 | 45.22 | 82.46 | 61.58 | 89.58 | 43.77 |
| Gaussian Noise | Uni | 73.7 | 87.88 | 56.11 | 97.37 | 55.99 | 96.32 | 59.85 | 95.55 | 61.41 | 94.28 |
| | Multi | 95.47 | 23.58 | 90.74 | 47.27 | 88.26 | 52.06 | 85.4 | 57.82 | 89.97 | 45.18 |
| Shot Noise | Uni | 76.81 | 85.35 | 58.25 | 97.29 | 58.1 | 96.28 | 60.3 | 96.1 | 63.36 | 93.75 |
| | Multi | 95.97 | 21.26 | 90.85 | 46.16 | 88.44 | 51.13 | 85.18 | 58.09 | 90.11 | 44.16 |
| Zoom Blur | Uni | 67.78 | 93.04 | 69.32 | 91.24 | 66.3 | 92.24 | 65.61 | 90.94 | 67.25 | 91.86 |
| | Multi | 92.35 | 40.41 | 90.93 | 42.52 | 87.93 | 49.05 | 84.46 | 59.27 | 88.92 | 47.81 |
| Snow | Uni | 85.35 | 74.35 | 79.24 | 85.5 | 75.88 | 87.71 | 76.94 | 75.55 | 79.35 | 80.78 |
| | Multi | 95.35 | 26.59 | 91.94 | 39.16 | 88.76 | 45.93 | 86.6 | 53.4 | 90.66 | 41.27 |
| Impulse Noise | Uni | 74.25 | 90.69 | 53.61 | 98.17 | 54.37 | 97.44 | 63.86 | 94.18 | 61.52 | 95.12 |
| | Multi | 95.75 | 22.7 | 90.05 | 50.09 | 87.73 | 53.49 | 86.06 | 54.91 | 89.9 | 45.3 |

extended dimensions preferentially provide additional clues that make the OOD samples more salient, excessive extended dimensions or the inclusion of input dimensions transformed from some specific uncommon corruptions may result in a greater degree of interference in the ID samples, leading to a slight declination of the performance.

### C.2.4 Full Results of CoVer with Variants Corrupted at Different Severity Levels.

In Figure 5(c), we report three representative variation trends of performance with an increasing level of the corruption severity. In Table 10 and Table 11, we present the full results of the proposed CoVer with 18 different types of corruptions at 5 severity levels. It is worth noting that the experimental results here are all based on the average of the model confidence scores measured on one original input and one extended corrupted input. Furthermore, we present the performance changing trends of our CoVer based on all 18 corruption styles in Figure 7. It can be seen that the performance of CoVer is more sensitive to the severity levels of corruptions compared to the number of extended representation dimensions analyzed in Appendix C.2.3. Specifically, as shown in Figure 7, some common corruption types from categories including weather (e.g., *Brightness* and *Fog*) and blur (e.g., *Motion blur* and *Defocus blur*) can achieve lower FPR95 values. We further observe that the commonality of these better-performing corruption types is that their perturbations to image features are more mild compared to other types (e.g., *Spatter* and *Elastic transform* from the digital class, *Impulse noise* from the noise category), and they generally do not excessively corrupt the semantic features. In Appendix C.4, we will more intuitively demonstrate the distinctions between these various types of corruptions through the visualizations of corrupted ID and OOD samples, illustrating the varying degrees of enhancement or attenuation they impart on the performance of CoVer.

Table 9: Comparison with different numbers of expanded representation dimensions at corruption severity level 1 and 2 based on CLIP-B/16. The ID data are ImageNet-1K. For each number of expansions, we provide two choices for the corruption types combination.

| Severity Level | Expanded Nums | Input types | OOD Dataset | | | | | | | | | |
| | | | iNaturalist | | SUN | | Places | | Textures | | Average | |
| | | | AUROC↑ | FPR95↓ | AUROC↑ | FPR95↓ | AUROC↑ | FPR95↓ | AUROC↑ | FPR95↓ | AUROC↑ | FPR95↓ |
| | 0 | **Original** | 94.61 | 30.95 | 92.57 | 37.57 | 89.77 | 44.65 | 86.10 | 57.77 | 90.76 | 42.73 |
| Level 1 | 1 | original + contrast | 94.60 | 30.26 | 92.77 | 37.24 | 89.99 | 43.93 | 87.08 | 55.11 | 91.11 | 41.64 |
| | | original + defocus blur | 94.80 | 29.26 | **93.57** | **31.87** | **90.68** | **39.19** | 87.74 | 52.22 | 91.70 | 38.13 |
| | 2 | original + contrast + brightness | 94.67 | 29.87 | 92.85 | 37.21 | 90.17 | 43.79 | 87.21 | 55.20 | 91.22 | 41.52 |
| | | original + defocus blur + motion blur | 94.93 | 28.15 | 93.49 | 32.17 | 90.65 | 39.53 | 87.58 | 53.19 | 91.66 | 38.26 |
| | 3 | original + contrast + brightness + saturate | 94.95 | 28.36 | 92.88 | 36.22 | 90.16 | 43.58 | 87.28 | 55.11 | 91.32 | 40.82 |
| | | original + defocus blur + motion blur + fog | 95.18 | 26.54 | 93.40 | 32.72 | 90.62 | 39.95 | **88.08** | **51.88** | 91.82 | **37.77** |
| | 4 | original + contrast + brightness + saturate + gaussion blur | 94.91 | 28.82 | 92.99 | 35.99 | 90.30 | 43.08 | 87.36 | 55.04 | 91.39 | 40.73 |
| | | original + defocus blur + motion blur + fog + gaussian blur | 94.98 | 27.92 | 93.29 | 33.13 | 90.55 | 40.47 | 87.87 | 52.71 | 91.67 | 38.56 |
| | 5 | original + contrast + brightness + saturate + gaussion blur + fog | 95.03 | 27.40 | 92.96 | 35.84 | 90.29 | 42.93 | 87.66 | 53.78 | 91.49 | 40.04 |
| | | original + defocus blur + motion blur + fog + gaussian blur + saturate | 95.18 | **26.49** | 93.33 | 32.89 | 90.57 | 40.27 | 87.91 | 52.8 | 91.75 | 38.11 |
| | 6 | original + contrast + brightness + saturate + gaussion blur + fog + motion blur | 95.12 | 26.65 | 93.09 | 34.78 | 90.40 | 41.84 | 87.70 | 53.60 | 91.58 | 39.22 |
| | | original + defocus blur + motion blur + fog + gaussian blur + saturate + brightness | **95.20** | 26.63 | 93.34 | 33.23 | 90.63 | 40.43 | 87.93 | 52.62 | 91.78 | 38.23 |
| Level 2 | 1 | original + brightness | 94.63 | 29.73 | 92.67 | 36.69 | 90.09 | 43.55 | 87.07 | 54.65 | 91.11 | 41.16 |
| | | original + defocus blur | 94.85 | 28.64 | **93.60** | 31.77 | **90.57** | 39.60 | 88.21 | 50.25 | 91.81 | 37.56 |
| | 2 | original + brightness + contrast | 94.91 | 28.57 | 92.92 | 36.53 | 90.18 | 43.57 | 87.81 | 53.33 | 91.45 | 40.50 |
| | | original + defocus blur + motion blur | 94.93 | 27.81 | 93.52 | **31.61** | 90.38 | 39.85 | 87.88 | 51.37 | 91.68 | 37.66 |
| | 3 | original + brightness + contrast + saturate | 95.70 | 24.90 | 92.83 | 36.67 | 90.10 | 44.22 | 88.01 | 52.82 | 91.66 | 39.65 |
| | | original + defocus blur + motion blur + gaussion blur | 94.44 | 29.86 | 93.23 | 33.15 | 90.03 | 40.99 | 87.70 | 52.15 | 91.35 | 39.04 |
| | 4 | original + brightness + contrast + saturate + fog | **95.74** | **23.68** | 92.83 | 36.66 | 90.11 | 44.07 | 88.36 | 51.06 | 91.76 | 38.87 |
| | | original + defocus blur + motion blur + gaussion blur + fog | 94.88 | 27.91 | 93.29 | 32.91 | 90.21 | 40.72 | 88.34 | 50.50 | 91.68 | 38.01 |
| | 5 | original + brightness + contrast + saturate + fog + gaussion blur | 95.66 | 24.14 | 93.13 | 34.98 | 90.32 | 42.45 | **88.63** | 50.12 | 91.94 | 37.92 |
| | | original + defocus blur + motion blur + gaussion blur + fog + saturate | 95.54 | 25.07 | 93.33 | 32.74 | 90.33 | 40.87 | 88.60 | 50.28 | 91.95 | **37.24** |
| | 6 | original + brightness + contrast + saturate + fog + gaussion blur + motion blur | 95.68 | 24.12 | 93.28 | 33.70 | 90.40 | 41.56 | 88.59 | 50.46 | **91.99** | 37.46 |
| | | original + defocus blur + motion blur + gaussion blur + fog + saturate + contrast | 95.43 | 25.45 | 93.28 | 33.21 | 90.25 | 41.23 | 88.62 | **50.09** | 91.90 | 37.49 |

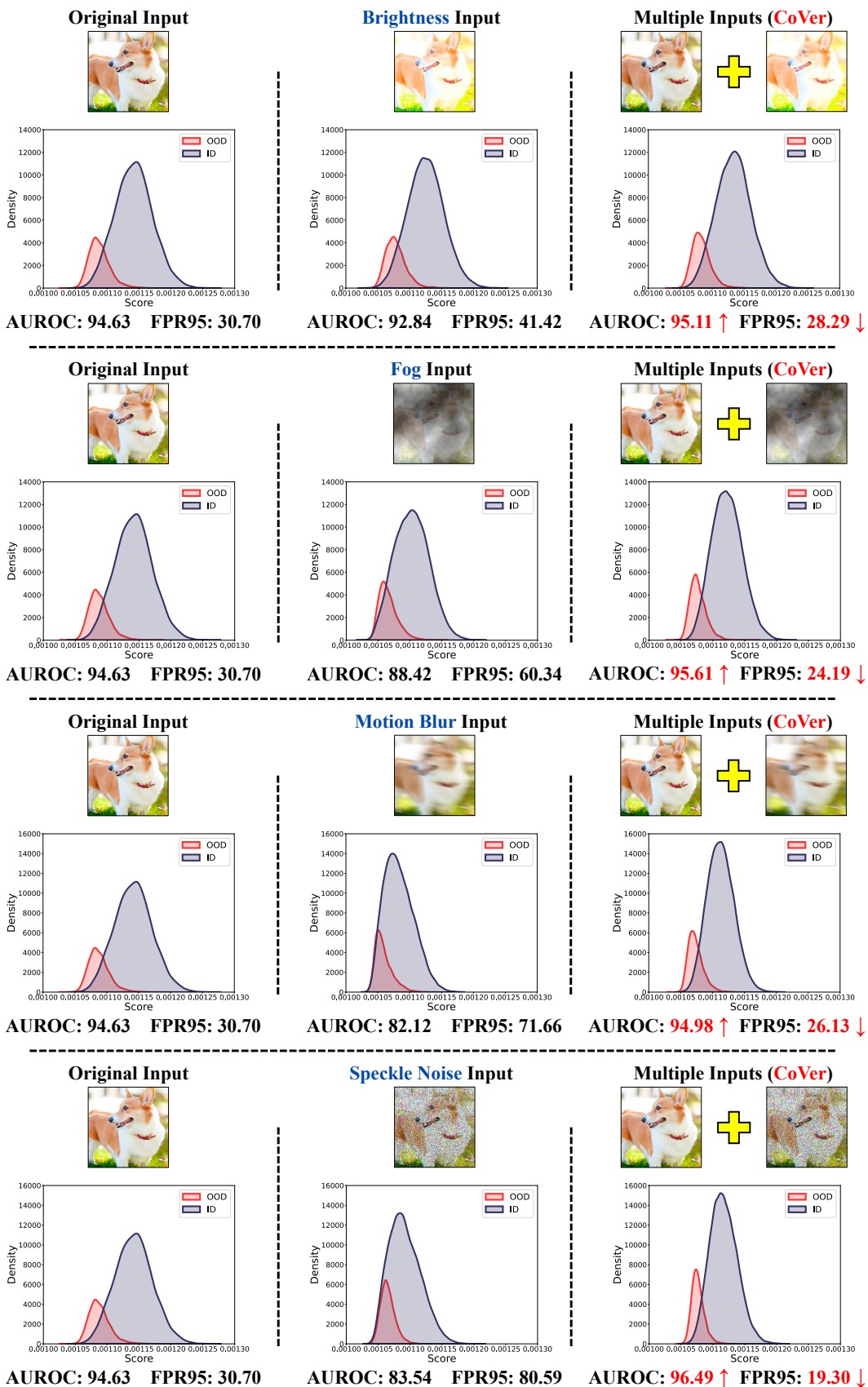

Figure 6: Comparison of scores distributions and detection results with different inputs for representation dimension expansion under various corruptions.

Table 10: Full results of CoVer under one extended input with 18 alternative types of corruptions at 5 severity levels based on CLIP-B/16. The ID data are ImageNet-1K.

| Corruption Type | Severity Level | OOD Dataset | | | | | | | | | |
| | | iNaturalist | | SUN | | Places | | Textures | | Average | |
| | | AUROC↑ | FPR95↓ | AUROC↑ | FPR95↓ | AUROC↑ | FPR95↓ | AUROC↑ | FPR95↓ | AUROC↑ | FPR95↓ |
|---|---|---|---|---|---|---|---|---|---|---|---|
| Brightness | 1 | 94.62 | 30.09 | 92.71 | 36.78 | **90.16** | 43.53 | 86.72 | 55.57 | 91.05 | 41.49 |
| | 2 | 94.63 | 29.73 | 92.67 | **36.69** | 90.09 | 43.55 | 87.07 | 54.65 | 91.11 | 41.16 |
| | 3 | 94.67 | 30.24 | 92.63 | 37.17 | 90.00 | 43.53 | 87.51 | 53.67 | 91.20 | 41.15 |
| | 4 | 94.88 | 29.52 | 92.67 | 37.06 | 90.04 | 43.86 | 87.89 | 52.61 | 91.37 | 40.76 |
| | 5 | **95.11** | **28.29** | **92.76** | 36.84 | 90.05 | **43.08** | **88.08** | **52.30** | **91.50** | **40.13** |
| Fog | 1 | 95.24 | 26.09 | 92.76 | 37.00 | 90.12 | 42.95 | 87.67 | 52.68 | 91.45 | 39.68 |
| | 2 | 95.42 | 25.20 | 92.80 | 36.71 | 90.13 | 43.54 | 88.00 | 51.28 | 91.59 | 39.18 |
| | 3 | 95.56 | **24.11** | 93.02 | 34.80 | **90.21** | 42.31 | **88.26** | 48.94 | 91.76 | 37.54 |
| | 4 | 95.49 | 24.68 | 93.04 | 34.60 | 90.15 | 41.64 | 88.17 | 49.17 | 91.71 | 37.52 |
| | 5 | **95.61** | 24.19 | **93.19** | **33.77** | 90.15 | 40.96 | 88.11 | **48.72** | **91.77** | **36.91** |
| Contrast | 1 | 94.60 | 30.26 | 92.77 | 37.24 | 89.99 | 43.93 | 87.08 | 55.11 | 91.11 | 41.64 |
| | 2 | 94.88 | 28.37 | 92.88 | 36.71 | 90.02 | 43.87 | 87.54 | 53.78 | 91.33 | 40.68 |
| | 3 | 95.46 | 25.55 | **93.04** | 35.94 | **90.11** | 42.90 | 88.57 | 50.23 | 91.80 | 38.65 |
| | 4 | 96.37 | 20.10 | 92.95 | **35.18** | 89.85 | 42.13 | **90.13** | 43.90 | **92.32** | **35.33** |
| | 5 | **96.41** | **19.60** | 92.46 | 36.05 | 89.17 | 44.07 | 89.05 | 48.07 | 91.77 | 36.95 |
| Motion Blur | 1 | 95.18 | 26.65 | 93.24 | 33.21 | **90.54** | 40.30 | 87.03 | 54.38 | 91.50 | 38.63 |
| | 2 | **95.34** | 26.04 | **93.41** | **32.15** | 90.44 | **40.28** | 87.17 | 53.35 | **91.59** | **37.95** |
| | 3 | 95.16 | 26.14 | 93.06 | 32.88 | 89.75 | 41.62 | 87.26 | 51.88 | 91.31 | 38.13 |
| | 4 | 94.97 | **26.03** | 92.14 | 37.97 | 88.58 | 47.20 | **87.26** | **51.12** | 90.74 | 40.58 |
| | 5 | 94.98 | 26.13 | 91.40 | 42.02 | 87.78 | 50.18 | 87.24 | 50.92 | 90.35 | 42.31 |
| Defocus Blur | 1 | 94.80 | 29.26 | 93.57 | 31.87 | **90.68** | 39.19 | 87.74 | 52.22 | 91.70 | 38.13 |
| | 2 | **94.85** | **28.64** | **93.60** | **31.77** | 90.57 | 39.60 | 88.21 | 50.25 | **91.81** | **37.56** |
| | 3 | 94.65 | 29.04 | 93.14 | 33.75 | 90.13 | 41.39 | **88.52** | **50.16** | 91.61 | 38.59 |
| | 4 | 94.25 | 30.52 | 92.66 | 36.40 | 89.66 | 42.88 | 88.49 | 49.40 | 91.27 | 39.80 |
| | 5 | 94.04 | 30.80 | 92.12 | 39.58 | 89.09 | 45.25 | 88.32 | 49.80 | 90.89 | 41.36 |
| Gaussian Blur | 1 | 94.66 | 30.49 | 92.97 | 35.25 | 90.39 | 41.49 | 86.90 | 55.05 | 91.23 | 40.57 |
| | 2 | **94.80** | **29.34** | **93.39** | **32.99** | **90.44** | **40.33** | 87.69 | 51.74 | **91.58** | **38.60** |
| | 3 | 94.41 | 30.95 | 93.16 | 33.84 | 90.08 | 41.50 | 87.99 | 51.38 | 91.41 | 39.42 |
| | 4 | 93.94 | 33.01 | 92.65 | 36.77 | 89.59 | 43.91 | **88.03** | 50.90 | 91.05 | 41.15 |
| | 5 | 93.52 | 33.3 | 91.59 | 42.03 | 88.64 | 47.39 | 88.01 | **50.25** | 90.44 | 43.24 |
| Spatter | 1 | **94.48** | **30.85** | 92.89 | 35.54 | **90.30** | 42.06 | **86.45** | **56.44** | **91.03** | **41.22** |
| | 2 | 94.25 | 32.70 | **93.00** | **35.24** | 90.22 | **41.99** | 85.46 | 58.69 | 90.73 | 42.16 |
| | 3 | 94.18 | 34.14 | 93.01 | 36.06 | 90.01 | 42.80 | 84.68 | 61.05 | 90.47 | 43.51 |
| | 4 | 92.25 | 44.55 | 92.39 | 39.15 | 89.66 | 44.66 | 84.29 | 61.74 | 89.65 | 47.52 |
| | 5 | 92.06 | 44.98 | 92.30 | 40.42 | 89.46 | 46.10 | 83.89 | 62.36 | 89.43 | 48.47 |
| Saturate | 1 | 95.14 | 27.49 | **92.75** | 36.52 | 90.00 | 43.86 | 86.74 | 56.31 | 91.16 | 41.05 |
| | 2 | **96.06** | **22.22** | 92.38 | 38.12 | 89.74 | 44.80 | 87.15 | 54.27 | **91.33** | **39.85** |
| | 3 | 94.47 | 31.40 | 92.68 | 37.04 | **90.07** | 43.75 | 86.59 | 56.24 | 90.95 | 42.11 |
| | 4 | 94.58 | 30.42 | 92.44 | 38.03 | 89.86 | 44.05 | 87.20 | 54.11 | 91.02 | 41.65 |
| | 5 | 94.24 | 31.27 | 92.11 | 38.41 | 89.90 | 43.08 | **87.23** | **53.37** | 90.87 | 41.53 |
| Elastic Transform | 1 | **94.71** | **29.15** | **92.69** | **36.08** | **90.09** | **42.86** | **87.12** | **54.79** | **91.15** | **40.72** |
| | 2 | 94.04 | 31.90 | 92.15 | 38.64 | 89.00 | 45.90 | 85.29 | 58.01 | 90.12 | 43.61 |
| | 3 | 93.81 | 34.98 | 92.17 | 39.52 | 89.42 | 45.50 | 84.66 | 61.08 | 90.02 | 45.27 |
| | 4 | 93.15 | 38.06 | 91.80 | 41.82 | 88.94 | 47.06 | 82.71 | 64.72 | 89.15 | 47.91 |
| | 5 | 90.57 | 53.15 | 91.12 | 45.97 | 88.52 | 49.36 | 77.98 | 72.84 | 87.05 | 55.33 |
| JPEG Compression | 1 | 93.36 | 38.72 | 92.65 | 37.15 | 90.08 | 43.37 | 87.11 | **55.43** | 90.80 | 43.67 |
| | 2 | **93.39** | **38.20** | 92.99 | 35.26 | 90.37 | 41.91 | 87.04 | 55.90 | 90.95 | 42.82 |
| | 3 | 93.26 | 38.71 | **93.06** | **34.75** | 90.37 | 41.36 | 87.14 | 55.50 | 90.96 | 42.58 |
| | 4 | 93.11 | 39.86 | 93.01 | 35.27 | 90.20 | 41.63 | 86.92 | 55.76 | 90.81 | 43.13 |
| | 5 | 93.08 | 39.52 | 92.95 | 36.21 | 89.97 | 42.17 | 86.27 | 57.34 | 90.57 | 43.81 |
| Pixelate | 1 | 94.17 | 33.18 | 92.07 | 39.90 | **89.59** | 45.36 | 86.50 | 57.43 | 90.58 | 43.97 |
| | 2 | 94.32 | 31.99 | **92.07** | **39.41** | 89.56 | 45.51 | **86.69** | **56.79** | 90.66 | 43.42 |
| | 3 | 94.59 | 29.88 | 91.79 | 40.42 | 89.39 | 45.65 | 85.90 | 59.66 | 90.42 | 43.90 |
| | 4 | **94.66** | **29.32** | 91.42 | 41.88 | 88.88 | 47.21 | 85.64 | 59.54 | 90.15 | 44.49 |
| | 5 | 94.56 | 29.86 | 91.79 | 39.89 | 89.11 | 45.89 | 85.86 | 58.48 | 90.33 | 43.53 |
| Speckle Noise | 1 | 94.36 | 33.35 | **92.40** | **38.66** | 89.68 | 45.02 | 87.16 | 55.50 | 90.90 | 43.13 |
| | 2 | 94.60 | 31.45 | 92.37 | 38.71 | 89.60 | 45.02 | **87.17** | **54.96** | **90.94** | **42.54** |
| | 3 | 95.44 | 26.28 | 92.03 | 41.94 | 89.27 | 47.41 | 86.60 | 56.37 | 90.84 | 43.00 |
| | 4 | 96.03 | 22.62 | 91.70 | 43.94 | 88.95 | 49.31 | 86.00 | 57.23 | 90.67 | 43.27 |
| | 5 | **96.49** | **19.30** | 91.51 | 44.35 | 88.92 | 49.75 | 85.44 | 56.95 | 90.59 | 42.59 |
| Glass Blur | 1 | 94.75 | 30.02 | 92.87 | 35.99 | **89.98** | 42.55 | **86.00** | 59.17 | **90.90** | 41.93 |
| | 2 | **94.79** | **29.74** | **92.95** | **35.19** | 89.89 | 42.51 | 85.70 | 58.92 | 90.83 | **41.59** |
| | 3 | 94.30 | 30.71 | 91.97 | 40.13 | 88.76 | 46.88 | 82.40 | 63.01 | 89.36 | 45.18 |
| | 4 | 93.79 | 32.69 | 91.94 | 40.11 | 88.78 | 46.54 | 81.75 | 63.74 | 89.06 | 45.77 |
| | 5 | 94.51 | 30.81 | 92.23 | 37.47 | 89.12 | 45.22 | 82.46 | 61.58 | 89.58 | 43.77 |
| Gaussian Noise | 1 | 94.12 | 34.13 | **92.23** | 39.59 | **89.65** | 45.07 | 86.95 | 55.76 | 90.74 | 43.64 |
| | 2 | 94.34 | 32.43 | 92.18 | **39.25** | 89.44 | 45.64 | 87.27 | **53.53** | 90.81 | 42.71 |
| | 3 | 94.84 | 28.69 | 92.02 | 39.78 | 89.11 | 45.74 | **87.29** | 53.71 | **90.82** | **41.98** |
| | 4 | 94.87 | 28.32 | 91.22 | 44.79 | 88.50 | 49.26 | 86.40 | 56.28 | 90.25 | 44.66 |
| | 5 | **95.47** | **23.58** | 90.74 | 47.27 | 88.26 | 52.06 | 85.40 | 57.82 | 89.97 | 45.18 |

Table 11: (Extension of Table 10) Full results of CoVer under one extended input with 18 alternative types of corruptions at 5 severity levels based on CLIP-B/16. The ID data are ImageNet-1K.

| Corruption Type | Severity Level | iNaturalist | | SUN | | Places | | Textures | | Average | |
|---|---|---|---|---|---|---|---|---|---|---|---|
| | | AUROC↑ | FPR95↓ | AUROC↑ | FPR95↓ | AUROC↑ | FPR95↓ | AUROC↑ | FPR95↓ | AUROC↑ | FPR95↓ |
| Shot Noise | 1 | 94.37 | 32.83 | **92.34** | 39.34 | **89.59** | 45.12 | 87.14 | 55.73 | 90.86 | 43.25 |
| | 2 | 94.82 | 29.23 | 92.29 | **38.92** | 89.49 | **44.90** | **87.18** | **55.21** | **90.95** | **42.07** |
| | 3 | 95.23 | 26.53 | 91.81 | 42.03 | 89.04 | 47.28 | 86.77 | 55.67 | 90.71 | 42.88 |
| | 4 | 95.80 | 22.83 | 91.08 | 46.01 | 88.56 | 50.64 | 85.74 | 57.64 | 90.30 | 44.28 |
| | 5 | **95.97** | **21.26** | 90.85 | 46.16 | 88.44 | 51.13 | 85.18 | 58.09 | 90.11 | 44.16 |
| Zoom Blur | 1 | **94.63** | **29.75** | **93.16** | **32.44** | **90.20** | **40.27** | **87.18** | **52.57** | **91.29** | **38.76** |
| | 2 | 94.08 | 33.05 | 92.70 | 35.13 | 89.62 | 42.70 | 87.00 | 53.03 | 90.85 | 40.98 |
| | 3 | 93.50 | 35.87 | 92.15 | 37.36 | 89.01 | 45.09 | 85.95 | 55.66 | 90.15 | 43.49 |
| | 4 | 92.95 | 37.98 | 91.65 | 39.51 | 88.48 | 46.92 | 85.63 | 57.04 | 89.68 | 45.36 |
| | 5 | 92.35 | 40.41 | 90.93 | 42.52 | 87.93 | 49.05 | 84.46 | 59.27 | 88.92 | 47.81 |
| Snow | 1 | 94.35 | 32.94 | **92.62** | **36.32** | **89.71** | **43.21** | 85.82 | 58.72 | 90.62 | 42.80 |
| | 2 | **94.63** | 30.66 | 92.23 | 37.93 | 89.18 | 44.87 | 86.15 | 56.47 | 90.55 | 42.48 |
| | 3 | 94.57 | **30.42** | 92.18 | 37.94 | 89.18 | 44.78 | 86.20 | 55.80 | 90.53 | 42.23 |
| | 4 | 94.59 | 30.74 | 91.80 | 39.69 | 88.79 | 46.04 | 86.35 | 55.16 | 90.38 | 42.91 |
| | 5 | 95.35 | 26.59 | 91.94 | 39.16 | 88.76 | 45.93 | **86.60** | **53.40** | **90.66** | **41.27** |
| Impulse Noise | 1 | 93.33 | 41.46 | **92.61** | **38.58** | **89.70** | **44.57** | 86.37 | 57.32 | 90.50 | 45.48 |
| | 2 | 94.24 | 33.91 | 91.94 | 43.03 | 89.11 | 47.87 | 86.09 | 56.26 | 90.34 | **45.27** |
| | 3 | 94.43 | 31.89 | 91.41 | 45.29 | 88.63 | 49.98 | 86.05 | 55.76 | 90.13 | 45.73 |
| | 4 | 95.11 | 26.64 | 90.73 | 47.75 | 88.04 | 52.06 | 86.01 | **54.56** | 89.97 | 45.25 |
| | 5 | **95.75** | **22.70** | 90.05 | 50.09 | 87.73 | 53.49 | 86.06 | 54.91 | 89.90 | 45.30 |

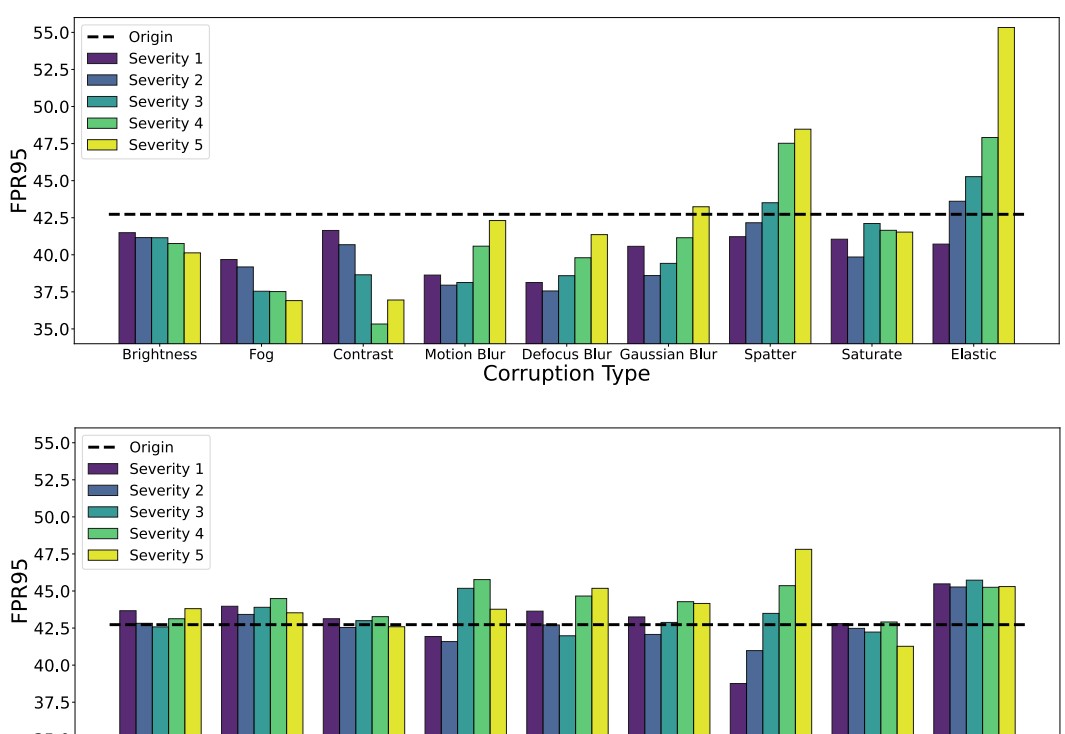

Figure 7: Performance variation trends of CoVer by varying severity levels under different types of corruptions.

## C.3 Exploration on the Formulation of CoVer

In Section 3.3, we discuss about the scheme of designing the OOD score with extra corrupted inputs:

$$S_{\text{CoVer}} = \mathbb{E}_{x \sim d(\mathcal{X}, \tilde{\mathcal{X}})}(S(x)), \quad d(\mathcal{X}, \tilde{\mathcal{X}}) := \{x, c(x) | x \in \mathcal{X}, c \in C\}, \tag{17}$$

The critical mechanism of $S_{\text{CoVer}}$ can be summarized as follows: $S_{\text{CoVer}}$ is an confidence expectation with respect to the OOD score estimated from natural input and its corrupted variant.

To have a closer look at our proposed CoVer, we further separately explain the implementation mechanism of CoVer at the input level and the output level. At the input level, CoVer is an extension of the input representation dimension, which introduces additional variant feature inputs into the model. On the other hand, CoVer is an extension of the confidence assessment dimension at the output level, which enables the integration of multi-dimensional confidence by averaging the confidence under every representation dimension. Such a design criterion leaves a further exploration on the specific formulation of CoVer. In other words, the realization of the estimated confidence score under each evaluation dimension can have multiple choices.

Table 12 shows the results of different implementation schemes for the CoVer score function. We also provide the original score function formulations of each method and their performance for comparison. All experiments in this table are evaluated on ImageNet-1K benchmark and all methods are based on CLIP-B/16. The results indicate that these scoring methods can achieve more comparable performances under the setting of the CoVer mode, as analyzed in Section 4.2,

Turning the attention to the implementation form of the proposed CoVer, we divide it into two categories: traditional scoring functions (from row 1 to row 4) and newly designed scoring functions based on negative textual logic (from row 5 to row 8). For traditional maximum-softmax and free energy scoring functions, we simply employ the expectation of them on the expanded multiple representation dimensions to realize our CoVer, as demonstrated in Eq. (17). For novel CLIPN and NegLabel scoring functions that exploit extra textual information, we apply the CoVer's mechanism to the calculation of the similarity between the image input and the negative text input. Formally it can be summarized as follows:

$$S_{\text{CoVer}}(\mathbf{x}) = \mathbb{E}_{\mathbf{x} \sim d(\mathcal{X}), \tilde{\mathbf{x}} \sim d(\mathcal{X}, \tilde{\mathcal{X}})} S^* \left( \text{sim}(\mathbf{x}, \mathbf{y}), \text{sim}(\tilde{\mathbf{x}}, \mathbf{y}^-) \right) \tag{18}$$

where $\mathbf{x} \sim d(\mathcal{X})$ is denoted as the input image sampled from original input space, and $\tilde{\mathbf{x}} \sim d(\mathcal{X}, \tilde{\mathcal{X}})$ refers to the input image sampled from expanded input space. $\mathbf{x}$ and $\tilde{\mathbf{x}}$ are involved in calculating the similarity to the positive textual concept $\mathbf{y}$ and negative textual concept $\mathbf{y}^-$, respectively. $S^*$ is the realization form of CLIPN and NegLabel, which can be found in row 6 and row 8 of Table 12.

Table 12: Comparison with the original and CoVer's modified form of different scoring functions based on CLIP-B/16. All experiments are evaluated on ImageNet-1K benchmark.

| Method | Mode | Score Function | OOD Dataset | | | | | | | | | | |
|---|---|---|---|---|---|---|---|---|---|---|---|---|
| | | | iNaturalist | | SUN | | Places | | Textures | | Average | |
| | | | AUROC↑ | FPR95↓ | AUROC↑ | FPR95↓ | AUROC↑ | FPR95↓ | AUROC↑ | FPR95↓ | AUROC↑ | FPR95↓ |
| MSP | Original | $\max_i \frac{e^{s_i(x)/\tau}}{\sum_{j=1}^{K} e^{s_j(x)/\tau}}$ | 94.61 | 30.95 | 92.57 | 37.57 | 89.77 | 44.65 | 86.10 | 57.77 | 90.76 | 42.73 |
| | CoVer | $\mathbb{E}_{x \sim d(\mathcal{X}, \tilde{\mathcal{X}})} \max_i \frac{e^{s_i(x)/\tau}}{\sum_{j=1}^{K} e^{s_j(x)/\tau}}$ | **95.98** | **22.55** | **93.42** | **32.85** | **90.27** | **40.71** | **90.14** | **43.39** | **92.45** | **34.88** |
| Energy | Original | $\log \sum_{j=1}^{K} e^{s_j(x)/T}$ | 85.54 | 80.49 | 84.21 | 78.75 | 84.81 | 72.29 | 66.63 | 92.89 | 80.30 | 81.11 |
| | CoVer | $\mathbb{E}_{x \sim d(\mathcal{X}, \tilde{\mathcal{X}})} \log \sum_{j=1}^{K} e^{s_j(x)/T}$ | **88.28** | **70.78** | **86.24** | **76.39** | **86.17** | **70.3** | **67.87** | **92.61** | **82.14** | **77.52** |
| CLIPN | Original | $\sum_{j=1}^{C} \frac{e^{s_{i,j}^{no}(x)/\tau}}{e^{s_{i,j}(x)/\tau} + e^{s_{i,j}^{no}(x)/\tau}} \cdot \frac{e^{s_{i,j}(x)/\tau}}{\sum_{k=1}^{C} e^{s_{i,k}(x)/\tau}}$ | 95.63 | **21.62** | 94.27 | 25.18 | 93.15 | 30.51 | 90.34 | 41.68 | 93.35 | 29.66 |
| | CoVer | $\mathbb{E}_{x \sim d(\mathcal{X}), \tilde{x} \sim d(\mathcal{X}, \tilde{\mathcal{X}})} \sum_{j=1}^{C} \frac{e^{s_{i,j}^{no}(x)/\tau}}{e^{s_{i,j}(x)/\tau} + e^{s_{i,j}^{no}(x)/\tau}} \cdot \frac{e^{s_{i,j}(x)/\tau}}{\sum_{k=1}^{C} e^{s_{i,k}(x)/\tau}}$ | 95.41 | 23.14 | **95.72** | **17.13** | **94.80** | **23.05** | 88.59 | **40.82** | **93.63** | **26.04** |
| NegLabel | Original | $\frac{\sum_{i=1}^{K} e^{s_i(x)/\tau}}{\sum_{i=1}^{K} e^{s_i(x)/\tau} + \sum_{j=1}^{M} e^{s_j^{neg}(x)/\tau}}$ | 99.49 | 1.93 | **95.46** | **20.95** | 91.58 | 36.45 | 89.89 | 45.12 | 94.10 | 26.11 |
| | CoVer | $\mathbb{E}_{x \sim d(\mathcal{X}), \tilde{x} \sim d(\mathcal{X}, \tilde{\mathcal{X}})} \frac{\sum_{i=1}^{K} e^{s_i(x)/\tau}}{\sum_{i=1}^{K} e^{s_i(x)/\tau} + \sum_{j=1}^{M} e^{s_j^{neg}(x)/\tau}}$ | **99.59** | **1.15** | 94.56 | 28.84 | **95.01** | **25.65** | **92.39** | 40.39 | **95.39** | **24.01** |

## C.4 Visualization Analysis

### C.4.1 Visualization of Corrupted ID and OOD Samples

In Figure 8, we provide the visualized examples under 18 different types of algorithmically generated corruptions which are officially introduced in [22]. Furthermore, there are 5 severity levels for each corruption style, ranging from inconsequential to devastating corruption of the original clean image as shown in Figure 9. By synthesizing these 18 types and 5 severity levels, we generate a comprehensive and diverse set of 90 distinct forms of corrupted inputs, which can be leveraged to enhance the dimensionality of input representation.

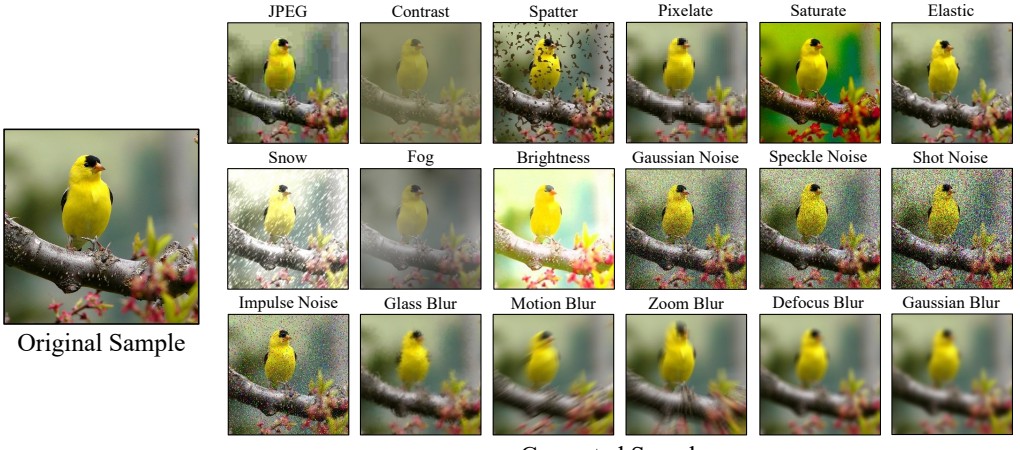

Figure 8: Visualization of an original sample and its corrupted instances under each corruption type officially defined in [22]

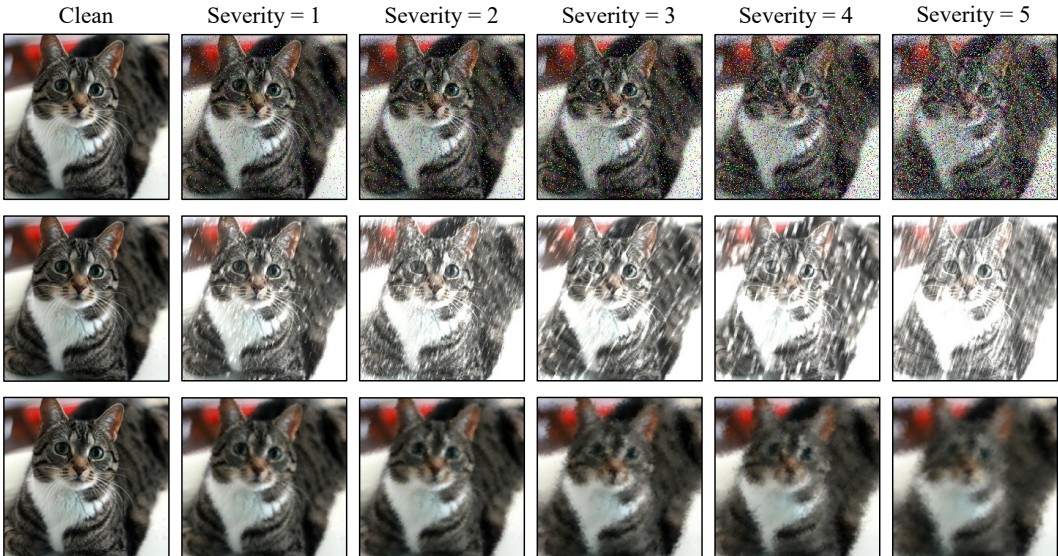

Figure 9: Visualization of varying severity levels, with *Impulse Noise*, *Snow*, and *Glass Blur* (all introduced in [22]) modestly to markedly corrupting the natural clean image.

### C.4.2 Comparison with Salient Maps and Corresponding Confidence Variations.

To better illustrate the effectiveness of CoVer, which primarily stems from the differing confidence variations between ID and OOD samples under identical corruption conditions, we provide some visualization results of ID and OOD images, as shown in Figure 10 and Figure 11. All the images are picked from the datasets in the ImageNet-1K OOD detection benchmark. Each subfigure shows the feature maps of the original image and its corrupted variants, including *Contrast* (Severity Level 4) and *Defocus Blur* (Severity Level 2). For comparison, we also provide the corresponding confidence variations between the original (red bar), the corrupted (blue bar), and the averaged (yellow bar) one. The confidence scores are based on the form of maximum-softmax scoring function given by CLIP-B/16. We continue to divide the data into four categories, denoted as confident ID data (refer to row 1 to row 2 in Figure 10), unconfident ID data (refer to row 3 to row 6 in Figure 10), overconfident OOD data (refer to row 1 to row 4 in Figure 11), and unconfident OOD data (refer to row 5 to row 6 in Figure 11). Here we focus on the differences between unconfident ID data and overconfident OOD to verify the analysis claimed in Section 3.2.

In Figure 10, it is obvious that ID images have more significant ID-semantic feature activations (see the foreground salient responses) due to the knowledge of the ID label space. Firstly, for confident ID data, the changes in confidence post-corruption can manifest as either minor or abrupt. For example, in the image in row 1, column 1 (ILSVRC2012_val_00020025), the corruption results in only a negligible loss of the semantics of the ID category present in the foreground. Conversely, for the image in row 2, column 1 (ILSVRC2012_val_00044407), the corruption enhances disturbances from non-semantic areas, leading to a loss of model confidence. Nevertheless, by averaging the original and corrupted confidence scores, the confident ID data remains stable within a higher confidence interval, demonstrating the superiority of CoVer's mechanism. Secondly, for unconfident ID data, due to the presence of ID semantic features, the model exhibits resilience in its confidence when subjected to corruption. For instance, the image in row 4, column 1 (ILSVRC2012_val_00022503) shows the model's resilience, as it continues to focus on the foreground regions belonging to the ID category despite various degrees of corruption, thus maintaining its confidence score unaffected. The same results can also be seen in row 3, column 1 (ILSVRC2012_val_00048997) and row 4, column 2 (ILSVRC2012_val_00020119).

In Figure 11, particularly in the case of overconfident OOD images, there is an excessive reaction to the corruption. This phenomenon is apparent because, unlike ID data, OOD images inherently lack semantic information about ID categories, leading to the disappearance of areas with high feature activation under the same type and severity level of corruption. For instance, the image in row 2, column 1 (f_formal_garden_00003688) demonstrates that regions highly responsive in their original state become irrelevant after corruption, resulting in the confidence mutation. This shift further confirms that the model's overconfidence in them is primarily due to misleading non-semantic features, such as textures and styles. Similar results can also be found in other OOD datasets, such as the image in row 3, column 1 (sun_arsrlxiznzlekfvg), and the image in row 4. column 2 (wrinkled_0070). However, the confidence of unconfident ID data, comparable to that of these overconfident OOD samples, remains stable. Consequently, by averaging the confidence scores before and after corruption, CoVer effectively captures the distinctions between OOD and ID data that initially overlap, thus enhancing the separability of ID and OOD distributions. Furthermore, for unconfident OOD data, due to their overall low relevance to ID semantics, the confidence scores consistently remain in a lower range (like the image in row 5, column 1, named 1b0ac86be7f53fd9058646315ed17269).

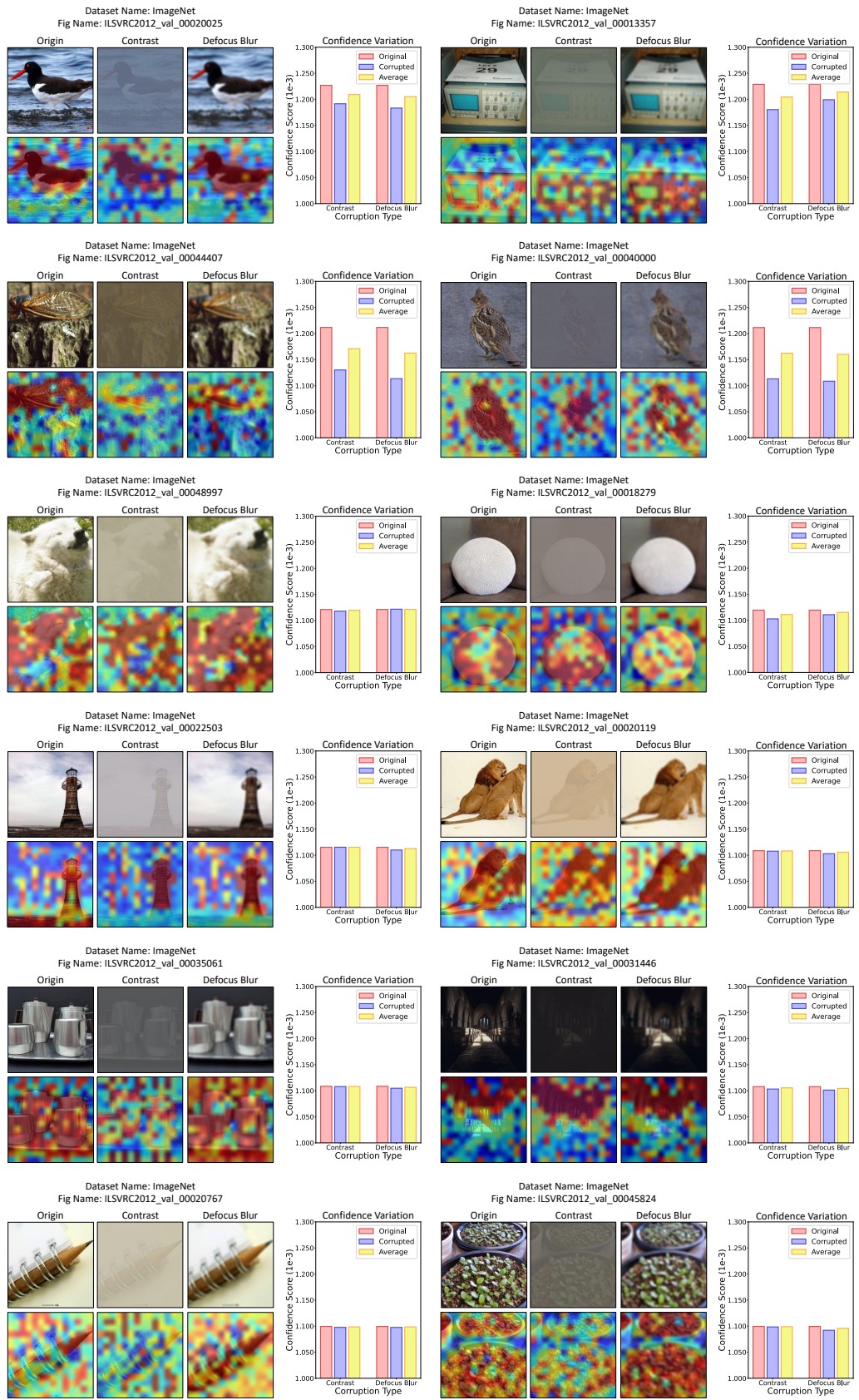

Figure 10: Case visualization of ID images. The left part of each subfigure contains the original image (with the dataset name and filename) and its corruptions with their feature maps. The right part shows the confidence variations corresponding to each corruption.

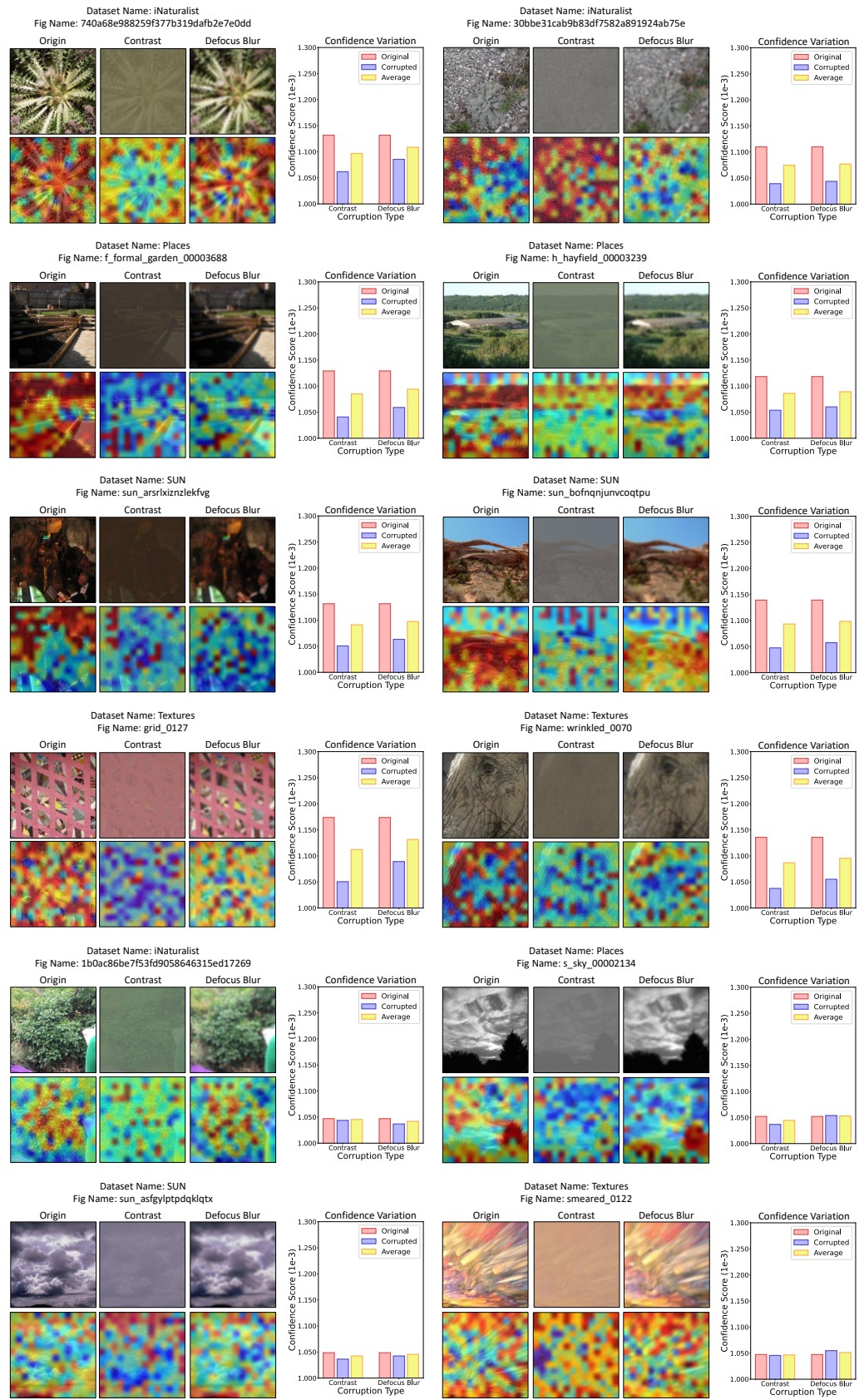

Figure 11: Case visualization of OOD images. The left part of each subfigure contains the original image (with the dataset name and filename) and its corruptions with their feature maps. The right part shows the confidence variations corresponding to each corruption.

# D Preliminary Statistical Analysis

In this section, we present the statistical implications with detailed definitions and assumptions. The primary objective is to show that CoVer can lead to an increase in the separability of the distributions of ID and OOD by introducing confidence expectations under the extended representation dimension. In the following parts, we first introduce the considering performance metric and some preliminary setups for the analyses.

**Metric.** The separability between ID and OOD data can be reflected by the $\mathrm{FPR}_\lambda$, which is the performance metric of an OOD detector, defined as follows,

$$\mathrm{FPR}_\lambda = F_{\mathrm{out}}\left(F_{\mathrm{in}}^{-1}(\lambda)\right) \tag{19}$$

where $F_{\mathrm{in}}$ and $F_{\mathrm{out}}$ represent the cumulative distribution functions (CDFs) corresponding to the confidence scores obtained by ID and OOD samples, respectively. $\lambda \in [0, 1]$ is denoted as the true positive rate (TPR), indicating the proportion of samples that are correctly classified as one of the ID categories. The metric $\mathrm{FPR}_\lambda$ quantifies the overlap degree between the scores that the OOD detector assigns to ID and OOD samples, with lower values indicative of superior performance.

**Preliminary setups.** Following previous works [32, 31], owing to the robust representational capabilities of pre-trained DNNs and the consistent alignment between cosine similarity scores and labels observed in CLIP-like models, we assume that the features extracted from DNNs or the cosine similarity scores in CLIP-like models approximately conform to a Gaussian Mixture Model (GMM) with equal class priors: $\left(\frac{1}{2}\mathcal{N}\left(\mu_{pi}, \sigma_{pi}\right) + \frac{1}{2}\mathcal{N}\left(\mu_{po}, \sigma_{po}\right)\right)$, where $\mu_{pi}$ and $\mu_{po}$ are the means of the ID and OOD distribution, while $\sigma_{pi}$ and $\sigma_{po}$ are the corresponding standard deviations.. Specifically, we use $\mathcal{D}_{\mathrm{ID}} = \mathcal{N}\left(\mu_{pi}, \sigma_{pi}\right)$ and $\mathcal{D}_{\mathrm{OOD}} = \mathcal{N}\left(\mu_{po}, \sigma_{po}\right)$ denote the ID marginal distribution and the OOD marginal distribution, respectively.

**Assumptions.** Refer to Figure 1 and Figure 6, the comparisons of score distributions obtained by different input modes, we can derive a series of assumptions about the variation relationships between $\mu_{pi}$ and $\mu_{po}$, and between $\sigma_{pi}$ and $\sigma_{po}$. Empirical exploration can be found in Figure 12.

**Assumption D.1** (Variation of $\mu_{pi}$ and $\mu_{po}$). *Let $\Delta\mu_{pi}$ and $\Delta\mu_{po}$ represent the changes in the means of ID and OOD distributions, respectively, after corruption and averaging. We assume that $|\Delta\mu_{pi}| > |\Delta\mu_{po}|$, resulting in a narrowing gap between $\mu_{pi}$ and $\mu_{po}$.*

This assumption is predicated on the observation that the means of ID distributions, $\mu_{pi}$, decrease more significantly under identical corruption levels compared to OOD distributions, $\mu_{po}$. The generally higher initial confidence scores of ID samples make their means more susceptible to substantial decreases (refer to the left panel of Figure 12). This reduction is greater than that experienced by OOD samples, thereby significantly narrowing the gap between $\mu_{pi}$ and $\mu_{po}$. For an intuitive example, the gap between the means of confident ID data (left panel of Figure 12) and overconfident OOD data (right-middle panel of Figure 12) is closer. This illustrates the pronounced impact of the averaging operation on ID distributions compared to OOD distributions.

**Assumption D.2** (Variation of $\sigma_{pi}$ and $\sigma_{po}$). *We define the changes in the variances of ID and OOD distributions as $\Delta\sigma_{pi}$ and $\Delta\sigma_{po}$, respectively. We postulate that the reduction in variance for ID distributions, $\Delta\sigma_{pi}$, is greater than that for OOD distributions, $\Delta\sigma_{po}$: $|\Delta\sigma_{pi}| > |\Delta\sigma_{po}|$.*

This assumption is supported by the observation that high-confidence ID samples, due to their higher initial confidence levels, experience larger and more abrupt drops in confidence upon corruption (see the left panel of Figure 12). Consequently, the averaging process post-corruption results in a significantly greater reduction in the variance $\sigma_{pi}$ in ID distributions compared to $\sigma_{po}$ in OOD distributions. This marked decrease in variability within ID confidence scores, relative to the OOD ones, underscores the efficacy of the averaging operation in dramatically stabilizing the ID confidence scores more than the adjustments observed in OOD confidence scores.

Given the preliminaries and assumptions above, we can derive the following extended lemma to demonstrate the superior performance of CoVer.

**Lemma D.3** (Declination of $\mathrm{FPR}_\lambda$). *Assuming the variation relationships between $\mu_{pi}$ and $\mu_{po}$, and between $\sigma_{pi}$ and $\sigma_{po}$, CoVer enables a lower $\mathrm{FPR}_\lambda$.*

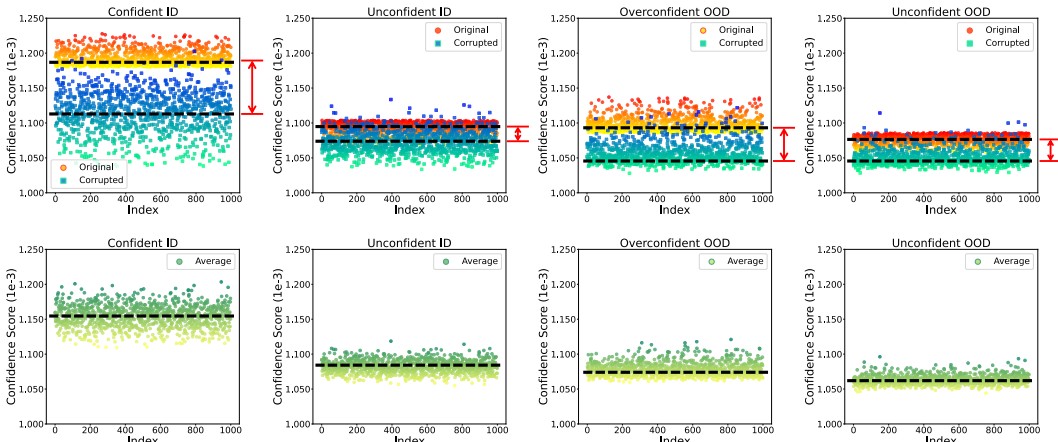

Figure 12: Empirical exploration to evidence the proposed assumptions. Based on Figure 2, we further present scatter maps of averaged confidence scores for comparison.

*Proof of Lemma D.3.* We aim to investigate the relationship between the $\mathrm{FPR}_\lambda$ metric and the variations in $\mu_{pi}$, $\mu_{po}$, $\sigma_{pi}$, and $\sigma_{po}$. The $\mathrm{FPR}_\lambda$ metric can be reformulated as follows:

$$
\begin{aligned}
\mathrm{FPR}_\lambda &= \Phi\left(\Phi^{-1}(\lambda; \mu_{pi}, \sigma_{pi}); \mu_{po}, \sigma_{po}\right) \\
&= \Phi\left(\mu_{pi} + \sigma_{pi} \cdot \Phi^{-1}(\lambda); \mu_{po}, \sigma_{po}\right) \\
&= \Phi\left(\frac{\mu_{pi} + \sigma_{pi} \cdot \Phi^{-1}(\lambda) - \mu_{po}}{\sigma_{po}}\right)
\end{aligned}
\tag{20}
$$

where $\Phi$ is he cumulative distribution function of the Gaussian distribution, and $\Phi^{-1}$ is its inverse function. Considering the differences before applying CoVer:

$$
\begin{aligned}
\Delta\mu &= \mu_{pi} - \mu_{po} \\
\Delta\sigma &= \sigma_{pi} - \sigma_{po}
\end{aligned}
\tag{21}
$$

With the assumptions that $\Delta\mu$ and $\Delta\sigma$ are affected by the averaging process in CoVer, we express these changes as:

$$
\begin{aligned}
\Delta\mu_{\text{new}} &= \mu'_{pi} - \mu'_{po} \\
\Delta\sigma_{\text{new}} &= \sigma'_{pi} - \sigma'_{po}
\end{aligned}
\tag{22}
$$

where $\mu'_{pi}$ and $\mu'_{po}$ are the new means post-CoVer, $\sigma'_{pi}$ and $\sigma'_{po}$ are the new variances post-CoVer. Given that $\Delta\mu_{\text{new}}$ is reduced and $\Delta\sigma_{\text{new}}$ indicates a significant contraction, particularly a reduction in $\sigma'_{pi}$ relative to $\sigma'_{po}$, the numerator in $(\mu_{pi} + \sigma_{pi} \cdot \Phi^{-1}(\lambda) - \mu_{po})/\sigma_{po}$, decreases while the denominator basically unchanged. This results in the argument of $\Phi$ becoming smaller. Since $\Phi$ is monotonically increasing, a decrease in the argument directly translates to a lower value of $\mathrm{FPR}_\lambda$, thereby reducing the probability of falsely classifying OOD samples as ID.

This analysis underscores the benefit of CoVer, particularly through its influence on expanding representation dimensions and optimizing the detection framework. By narrowing the gap between the mean scores and significantly reducing the variance in ID distributions relative to OOD, CoVer enhances the model's discriminative capability and improves its robustness, ultimately leading to more reliable OOD detection.

# E  Future Analysis

## E.1  Discussion about Extra Runtime

Considering the addition of corrupted inputs, CoVer would result in non-negligible additional runtime. If there are $N$ expanded dimensions, it will take $N$ times the duration of a single input to implement CoVer. However, our CoVer is only applied in the inference phase of OOD detection, and it is generally fast, as shown in Table 13. We believe that the performance improvements offered by CoVer are well-worth the extra few minutes of runtime.

Table 13: Inference runtime of a single input using a single RTX 3090 GPU based on CLIP-B/16.

| Dataset | Type | Inference time (s) |
|---|---|---|
| ImageNet | ID | 121 (±11) |
| iNaturalist | OOD | 45 (±15) |
| SUN | OOD | 41 (±12) |
| Places | OOD | 33 (±8) |
| Textures | OOD | 16 (±1) |

## E.2 Exploring the Impact of Corruption Types and Severity Levels

Appropriate types of corruptions and corresponding severity levels are crucial for the optimization of our CoVer. In all experiments, we use the SVHN dataset as the validation set to select the most effective corruption types for each method. Specific examples of selections are provided in Table 14. For the types of corrupted inputs and their corresponding severity levels, we have conducted related explorations (e.g., Tables 10 and 11, Figures 6 and 7 of our original submission) for performance references. Some specific corruptions (e.g., Brightness, Fog, Contrast, Motion Blur, Defocus Blur) can generally improve the OOD detection performance, as the corruptions are mainly on the non-semantic level of the input, instead of damaging the semantic features too much like the other types. Empirically, refer to Table 15, we can use the same type of corruption as the expanded input (e.g., here Brightness with severity 1 used) to perform better than the original version. This provides the verification of the previous intuition about the general guidance for choosing appropriate corruption types, and understanding dimension expansion for OOD detection.

Table 14: Examples of corruption types selections in the utilized validation set SVHN based on ResNet50

| Selected? | $D_{in}$ | $D_{val}$ | Method | Expanded Type | AUROC↑ | FPR95↓ |
|---|---|---|---|---|---|---|
| | ImageNet-1K | SVHN | MSP | / | 97.45 | 13.80 |
| ✓ | ImageNet-1K | SVHN | MSP | Brightness | **98.81** | **6.36** |
| ✓ | ImageNet-1K | SVHN | MSP | Fog | **98.83** | **6.52** |
| ✓ | ImageNet-1K | SVHN | MSP | Motion Blur | **98.62** | **7.69** |
| ✗ | ImageNet-1K | SVHN | MSP | Snow | 89.43 | 56.47 |
| ✗ | ImageNet-1K | SVHN | MSP | Impulse Noise | 94.71 | 29.74 |
| ✗ | ImageNet-1K | SVHN | MSP | Spatter | 95.87 | 28.03 |

Table 15: CoVer combined with each method using the same expanded corruption type.

| Architecture | $D_{in}$ | Method | Expanded Type | AUROC↑ | FPR95↓ |
|---|---|---|---|---|---|
| ResNet50 | ImageNet-1K | ReAct | / | 92.95 | 31.43 |
| ResNet50 | ImageNet-1K | ReAct + CoVer | Brightness(1) | **93.94** | **28.10** |
| ResNet50 | ImageNet-1K | DICE | / | 90.77 | 34.75 |
| ResNet50 | ImageNet-1K | DICE + CoVer | Brightness(1) | **91.96** | **31.66** |
| ResNet50 | ImageNet-1K | ASH-B | / | 90.91 | 39.04 |
| ResNet50 | ImageNet-1K | ASH-B + CoVer | Brightness(1) | **92.24** | **30.55** |
| CLIP-B/16 | ImageNet-1K | MCM | / | 90.76 | 42.73 |
| CLIP-B/16 | ImageNet-1K | MCM + CoVer | Brightness(1) | **91.05** | **41.49** |
| CLIP-B/16 | ImageNet-1K | CLIPN | / | 93.35 | 29.66 |
| CLIP-B/16 | ImageNet-1K | CLIPN + CoVer | Brightness(1) | **93.47** | **27.82** |
| CLIP-B/16 | ImageNet-1K | NegLabel | / | 94.10 | 26.11 |
| CLIP-B/16 | ImageNet-1K | NegLabel + CoVer | Brightness(1) | **95.15** | **24.99** |

### E.3 Exploration on the harder OOD dataset

NINCO [4] has proposed three OOD datasets with no categorical contamination which include NINCO, OOD unit-tests, and NINCO popular OOD datasets subsamples, which are demonstrated to be harder than common OOD detection benchmarks. Here, we evaluate the effectiveness of CoVer on these datasets in 16. The results demonstrate that CoVer, when combined with ASH, consistently achieves better performance across the three NINCO OOD datasets.

Table 16: The overall results of CoVer on three NINCO OOD datasets without leveraging VLMs/CLIP. The ID data are ImageNet-1K.

| Architecture | $D_{in}$ | $D_{out}$ | Method | AUROC↑ | FPR95↓ |
|---|---|---|---|---|---|
| ResNet50 | ImageNet-1K | NINCO | ASH | 82.26 | 69.22 |
| | | NINCO | AHS + CoVer | **82.80** | **68.59** |
| | | NINCO unit-tests | ASH | 99.13 | 4.85 |
| | | NINCO unit-tests | AHS + CoVer | **99.49** | **2.12** |
| | | NINCO subsamples | ASH | 82.07 | 56.10 |
| | | NINCO subsamples | AHS + CoVer | **82.67** | **54.44** |

### E.4 Comparison with other Competitive Methods

**Comparison with NNGuide and MaxLogit.** To provide a comprehensive comparisons, we have conducted comparison experiments with NNGuide [39] and MaxLogit [21] to enrich our analysis in Table 17. First, our experimental results show that CoVer outperforms these competitive post-hoc methods on the ResNet50 architecture. Second, the performance of these post-doc methods, especially NNGuide, encounter significant drop when conducted on CLIP-B/16 architecture. We believe the reason for the poor performance is the difference in training data. Many pos-hoc methods are designed on ImageNet pre-trained networks, where only ID data are used during training. In contrast, when training CLIP, both ID datas and OOD datas are used. This leads to different activations of OOD data. Another reason is that the pos-hoc method relies heavily on the choice of hyperparameters. The hyperparameters of NNGuide need to be re-selected on different models. Despite these issues, our CoVer can still perform better than these methods. Furthermore, we also combine our CoVer with MaxLogit and NNGuide and report the results in Table 18, which further demonstrates the effectiveness and compatibility of our method.

Table 17: Comparison with NNGuide and MaxLogit based on ResNet50 and CLIP-B/16. The ID data are ImageNet-1K.

| Architecture | Method | OOD Dataset | | | | | | | | | |
|---|---|---|---|---|---|---|---|---|---|---|---|
| | | iNaturalist | | SUN | | Places | | Textures | | Average | |
| | | AUROC↑ | FPR95↓ | AUROC↑ | FPR95↓ | AUROC↑ | FPR95↓ | AUROC↑ | FPR95↓ | AUROC↑ | FPR95↓ |
| ResNet50 | MaxLogit | 91.93 | 50.91 | 86.59 | 59.87 | 84.18 | 65.68 | 86.40 | 54.36 | 87.07 | 57.70 |
| | NNGuide(k=1) | 93.13 | 34.06 | 90.41 | 38.86 | 88.06 | 47.46 | 91.67 | 29.89 | 90.82 | 37.57 |
| | NNGuide(k=10) | 94.33 | 29.27 | 91.23 | 36.4 | 88.71 | 46.2 | 92.93 | 26.31 | 91.80 | 34.55 |
| | NNGuide(k=100) | 95.10 | 26.06 | 91.44 | 36.86 | 88.63 | 47.64 | **93.61** | **24.17** | 92.19 | 33.68 |
| | CoVer (ours) | **97.14** | **14.04** | **94.12** | **25.77** | **91.05** | **35.93** | 91.93 | 30.39 | **93.56** | **26.53** |
| CLIP-B/16 | MaxLogit | 89.31 | 61.66 | 87.43 | 64.39 | 85.95 | 63.67 | 71.68 | 86.61 | 83.59 | 69.08 |
| | NNGuide(k=1) | 65.06 | 99.38 | 68.56 | 97.27 | 72.19 | 93.51 | 66.06 | 98.49 | 67.97 | 97.16 |
| | NNGuide(k=10) | 60.98 | 99.68 | 68.06 | 98.06 | 71.65 | 94.83 | 62.61 | 98.99 | 65.83 | 97.89 |
| | NNGuide(k=100) | 51.34 | 99.85 | 64.84 | 98.83 | 68.74 | 96.49 | 53.26 | 99.63 | 59.54 | 98.70 |
| | CoVer (ours) | **95.98** | **22.55** | **93.42** | **32.85** | **90.27** | **40.71** | **90.14** | **43.39** | **92.45** | **34.88** |

Table 18: Compatibility experiments of CoVer combined with NNGuide and MaxLogit based on ResNet50 and CLIP-B/16. The ID data are ImageNet-1K.

| Architecture | Method | OOD Dataset | | | | | | | | | |
|---|---|---|---|---|---|---|---|---|---|---|---|
| | | iNaturalist | | SUN | | Places | | Textures | | Average | |
| | | AUROC↑ | FPR95↓ | AUROC↑ | FPR95↓ | AUROC↑ | FPR95↓ | AUROC↑ | FPR95↓ | AUROC↑ | FPR95↓ |
| **CLIP-B/16** | MaxLogit | 89.31 | 61.66 | 87.43 | 64.39 | 85.95 | 63.67 | 71.68 | 86.61 | 83.59 | 69.08 |
| | **MaxLogit+CoVer** | **91.78** | **49.93** | **89.20** | **59.64** | **87.89** | **59.15** | **74.01** | **84.50** | **85.72** | **63.31** |
| **ResNet50** | MaxLogit | 91.13 | 50.91 | 86.59 | 59.87 | 84.18 | 65.68 | 86.40 | 54.36 | 87.07 | 57.70 |
| | **MaxLogit+CoVer** | **92.85** | **42.19** | **87.19** | **58.17** | **84.97** | **63.04** | **86.59** | **54.10** | **87.90** | **54.38** |
| | NNGuide(k=1) | 93.13 | 34.06 | 90.41 | 38.86 | 88.06 | 47.46 | 91.67 | 29.89 | 90.82 | 37.57 |
| | **NNGuide (k=1)+CoVer** | **94.98** | **25.16** | **91.17** | **36.51** | **88.82** | **44.52** | **91.91** | **29.24** | **91.72** | **33.86** |
| | NNGuide(k=10) | 94.33 | 29.27 | 91.23 | 36.4 | 88.71 | 46.20 | 92.93 | 26.31 | 91.80 | 34.55 |
| | **NNGuide (k=10)+CoVer** | **95.84** | **21.61** | **91.91** | **34.45** | **89.38** | **43.42** | **93.12** | **25.80** | **92.56** | **31.32** |
| | NNGuide(k=100) | 95.10 | 26.06 | 91.44 | 36.86 | 88.63 | 47.64 | 93.61 | 24.17 | 92.19 | 33.68 |
| | **NNGuide (k=100)+CoVer** | **96.42** | **19.46** | **92.20** | **34.43** | **89.39** | **44.34** | **93.79** | **23.58** | **92.95** | **30.45** |

**Comparison with Watermarking.** In addition, Watermarking [54] is another competitive methods needed to be considered. We have added the analysis about the Watermarking method with our CoVer in the following two aspects.

Conceptually, we have noticed that Watermarking utilizes a well-trained mask to help the original images be distinguishable from the OOD data. However, Watermarking is still trying to excavate the useful feature representation in a single-input perspective. In contrast, the critical distinguishable point and also the advantage of our CoVer method lies in input dimension expansion with the corrupt variants, which instead provide a extra dimension to explore the confidence mutation to better identify the OOD samples.

Experimentally, we have conducted the comparison and report the results in Table 19. The results show that, on the one hand, training an optimized watermarking for effectively distinguishing between ID and OOD samples is a time-consuming process. On the other hand, CoVer achieves this by introducing corrupted inputs to capture the confidence variations between ID and OOD data during the test phase, which is simpler, faster, and more effective.

Table 19: Comparison with competitive OOD detection method Watermark based on ResNet50. The ID data are ImageNet-1K.

| Architecture | Method | OOD Dataset | | | | | | | | | | Runtime |
| | | iNaturalist | | SUN | | Places | | Textures | | Average | | |
| | | AUROC↑ | FPR95↓ | AUROC↑ | FPR95↓ | AUROC↑ | FPR95↓ | AUROC↑ | FPR95↓ | AUROC↑ | FPR95↓ | |
| ResNet50 | Watermark | 80.31 | 74.45 | 79.21 | 73.27 | 79.78 | 71.43 | 80.44 | 67.53 | 79.94 | 71.67 | 3 h/epoch |
| | **MSP+CoVer** | **90.81** | **44.90** | **82.51** | **66.38** | **81.57** | **69.34** | **81.00** | **65.43** | **83.97** | **61.41** | **10 mins** |

**Comparison with Data Depth, Information Projection, and Isolation Forest.** We have also conducted comparison experiments between our CoVer and baselines methods from data depths, information projections, and isolation forest, as detailed in Table 20.

Due to the large scale of the ImageNet training set, we sampled 50 samples per class to construct a subset from the training data to represent the training distribution, as recommended by the similar work named NNGuide [39]. For data depths, we reimplemented APPROVED [41] for comparison. For information projections, we reproduced REFEREE [40] for comparison. For Isolation Forest, we use logits as the input to detect the anomaly logits in ID and OOD samples.

The results indicate that AD and textual OOD detection methods, such as Data Depth and Information Projection, may not suit for visual OOD detection tasks, a view also mentioned in related surveys [8, 9]. Similarly, classical ML methods for AD, such as Isolation Forest, seem to be failed to excavate discriminative representations when applied to image OOD detection. However, since these methods are insightful in distinguishing the outliers, we believe it is worth further efforts in the future to adopt the critical intuition into the OOD detection problem.

Table 20: Comparison with competitive anomaly detection and textual OOD detection baselines based on ResNet50. The selected methods' types are Data Depth, Information Projection and Isolation Forest, respectively. The ID data are ImageNet-1K.

| Architecture | Method | OOD Dataset | | | | | | | | | |
| | | iNaturalist | | SUN | | Places | | Textures | | Average | |
| | | AUROC↑ | FPR95↓ | AUROC↑ | FPR95↓ | AUROC↑ | FPR95↓ | AUROC↑ | FPR95↓ | AUROC↑ | FPR95↓ |
| ResNet50 | APPROVED | 53.25 | 95.90 | 60.01 | 94.63 | 56.77 | 94.26 | 70.21 | 78.30 | 60.06 | 90.77 |
| | REFEREE | 79.77 | 94.76 | 73.28 | 96.91 | 72.9 | 96.80 | 74.01 | 94.08 | 74.99 | 95.64 |
| | Isolation Forest | 70.76 | 85.94 | 59.55 | 94.78 | 60.27 | 93.91 | 65.89 | 81.37 | 64.12 | 89.00 |
| | **MSP+CoVer** | **90.81** | **44.90** | **82.51** | **66.38** | **81.57** | **69.34** | **81.00** | **65.43** | **83.97** | **61.41** |

### E.5 Compatibility with each DNN-based mehtods

It is worth to note that, in Table 1, we only reported the results of CoVer combined with ASH because it best demonstrates the excellence of CoVer. In Table 3, we also show the results of CoVer combined with DICE and ReAct, and CoVer can also provide performance gains for them. Here in Table 21, we further report the comparison of CoVer combined with each mentioned DNN-based methods (adding MSP, ODIN, and Energy score), which strongly demonstrates its superiority.

Table 21: Compatibility experiments of CoVer combined with each mentioned DNN-based OOD detection method. The ID data are ImageNet-1K.

| Architecture | Method | OOD Dataset | | | | | | | | | |
|---|---|---|---|---|---|---|---|---|---|---|---|
| | | iNaturalist | | SUN | | Places | | Textures | | Average | |
| | | AUROC↑ | FPR95↓ | AUROC↑ | FPR95↓ | AUROC↑ | FPR95↓ | AUROC↑ | FPR95↓ | AUROC↑ | FPR95↓ |
| ResNet50 | MSP | 87.74 | 54.99 | 80.86 | 70.83 | 79.76 | 73.99 | 79.61 | 68.00 | 81.99 | 66.95 |
| | **MSP+CoVer** | **90.81** | **44.49** | **82.51** | **66.38** | **81.57** | **69.34** | **81.00** | **65.43** | **83.97** | **61.41** |
| | ODIN | 91.37 | 41.57 | 86.89 | 53.97 | 84.44 | 62.15 | 87.57 | 45.53 | 87.57 | 50.80 |
| | **ODIN+CoVer** | **93.66** | **31.56** | **88.14** | **51.47** | **85.98** | **58.69** | **87.97** | **44.77** | **88.94** | **46.62** |
| | Energy score | 89.95 | 55.72 | 85.89 | 59.26 | 82.86 | 64.92 | 85.99 | 53.72 | 86.17 | 58.41 |
| | **Energy+CoVer** | **92.23** | **46.67** | **87.42** | **56.50** | **84.98** | **63.16** | **86.99** | **51.70** | **87.91** | **54.51** |
| | ReAct | 96.22 | 20.38 | 94.20 | 24.20 | 91.58 | 33.85 | 89.80 | 47.30 | 92.95 | 31.43 |
| | **ReAct+CoVer** | **97.58** | **13.35** | **95.7** | **18.91** | **93.08** | **29.02** | **91.55** | **40.74** | **94.48** | **25.51** |
| | DICE | 94.49 | 25.63 | 90.83 | 35.15 | 87.48 | 46.49 | 90.30 | 31.72 | 90.77 | 34.75 |
| | **DICE+CoVer** | **96.8** | **16.56** | **93.53** | **28.52** | **90.00** | **40.54** | **91.14** | **31.15** | **92.87** | **29.19** |
| | ASH-B | 94.25 | 28.95 | 90.32 | 40.21 | 87.52 | 49.52 | 91.53 | 33.48 | 90.91 | 39.04 |
| | **ASH-B+CoVer** | **97.14** | **14.04** | **94.12** | **25.77** | **91.05** | **35.93** | **91.93** | **30.39** | **93.56** | **26.53** |

