# OpenReview forum: "What If the Input is Expanded in OOD Detection?"
_NeurIPS.cc/2024/Conference — NeurIPS 2024 poster_

### Official Review · Reviewer_hcf7 · 2024-07-08

**Soundness:** 3
**Presentation:** 4
**Contribution:** 3
**Rating:** 8
**Confidence:** 5

**Summary:**

In previous OOD detection methods, extracting discriminative information from OOD data relative to ID data is challenging with the representation of a single input. This paper provides a novel perspective for out-of-distribution (OOD) detection by leveraging multiple types of corruptions to expand the original single input space. Utilizing the interesting phenomenon of confidence mutation, the authors introduce a new scoring method termed CoVer, which averages the confidence scores measured from multiple input spaces, achieving better separability between ID and OOD distributions. Extensive experimentation underscores that the CoVer approach not only outperforms previous methods in both DNN-based and VLM-based OOD detection benchmarks, but also exhibits commendable compatibility when combined with different OOD detection methods. Moreover, the CoVer technique demonstrates exemplary adaptability across diverse VLM architectures.

**Strengths:**

1. Well written and technically sound. This paper is well organized and written. Both the motivation and the main contributions of the proposed method are easy to follow. Sufficient experiments have been conducted to demonstrate the effectiveness of the proposed method.

2.Novelty. The challenges of the previous methods are well summarized in OOD detection, i.e., single input constrains the representation ability for detection. The proposed method expands the input space to aware the ID and OOD distribution effectively.

3. Extensive experiments and ablations. The paper conducts extensive experiments, including OOD detection using different backbones, hard OOD detection, and various aspects of ablation studies. The performance improvements strongly verify the effectiveness of the proposed method.

4. Clear justification and theoretical analysis. The paper provides a clear and intuitive justification for the proposed method by a thorough theoretical analysis. This analysis clearly elucidates the mechanisms beyond CoVer, making the paper convincing and compelling.

**Weaknesses:**

1.Lack explaination and comparison. Different types of corruptions are used in the method, which is similar to the idea of Watermarking (Wang, et al. 2022), where a generic watermark is trained to distinguish between ID and OOD data. The advantages are not deeply analyzed.

2.Less clarification. The CoVer approach is flexible and compatible and can be combined with different methods to achieve performance gains. The combination is not clearly presented for different methods.

**Questions:**

1.Watermaking[1] is a very popular and SOTA method for OOD detection. Compared with watermarking and so on, what are the advantages of using corrupted images to expand the input space. Since the corrupted images are the important data in the proposed method, it is necessary to present the advantages for OOD with corrupted images more clearly.
[1]Wang Q, Liu F, Zhang Y, et al. Watermarking for out-of-distribution detection. Advances in Neural Information Processing Systems, 2022, 35.

2. The proposed method can enhance most of the OOD methods and improve their performance. However, these methods usually designed with different techniques, such as fine-tuning and zero-shot methods. Hence, I wonder whether there exsit differences to improve these methods with CoVer and how to guarantee the effectiveness when combine CoVer with these methods.

**Limitations:**

Yes

---

> ### Author Rebuttal · Authors · 2024-08-07
>
> Thank you for your time devoted to reviewing this paper and your constructive suggestions. Here are our detailed replies to your questions.
>
> > **W1, Q1:** Lack explaination and comparison. Different types of corruptions are used in the method, which is similar to the idea of Watermarking, where a generic watermark is trained to distinguish between ID and OOD data. The advantages are not deeply analyzed. Watermaking is a very popular and SOTA method for OOD detection. Compared with watermarking and so on, what are the advantages of using corrupted images to expand the input space. Since the corrupted images are the important data in the proposed method, it is necessary to present the advantages for OOD with corrupted images more clearly.
>
> Thank you for your constructive comments. **We have added the analysis about the Watermarking method with our CoVer in the following two aspects.**
>
> Conceptually, we have noticted that Watermarking utilizes a well-trained mask to help the original images be distinguishable from the OOD data. However, **Watermarking is still trying to excavate the useful feature representation in a single-input perspective**. In contrast, the critical distinguishable point and also the advantage of our CoVer method lies in input dimension expansion with the corrupt variants, which instead provide a extra dimension to explore the confidence mutation to better identify the OOD samples.
>
> Experimentally, we have conducted the comparison and report the results in **Table 9 in attached PDF**. The results show that, on the one hand, training an optimized watermarking for effectively distinguishing between ID and OOD samples is a **time-consuming process**. On the other hand, CoVer achieves this by introducing corrupted inputs to capture the confidence variations between ID and OOD data during the test phase, which is **simpler**, **faster**, and **more effective**. We will add this part in our revision and discuss them in detail in our appendix.
>
> > **W2, Q2:** Less clarification. The CoVer approach is flexible and compatible and can be combined with different methods to achieve performance gains. The combination is not clearly presented for different methods. The proposed method can enhance most of the OOD methods and improve their performance. However, these methods usually designed with different techniques, such as fine-tuning and zero-shot methods. Hence, I wonder whether there exsit differences to improve these methods with CoVer and how to guarantee the effectiveness when combine CoVer with these methods.
>
> Thank you for your thoughtful question. The fine-tuning methods designed for OOD detection focus on **utilizing extra auxiliary outliers** to regularize the model to be better aware of OOD data, while the zero-shot methods are focusing on **scoring function design** of excavating the original distinguisable feature that can better represent the differences. Although their focus are conceptually different, **our CoVer provides the similar enhancement as the adapation is on the input side** (i.e., adding the corrupted input dimension for score averaging). It sheds new light on leveraging raw input in an extra dimension, **but is not bound to either fine-tuning or zero-shot framework** that targets on utlizing or enhancing the single-input representativeness for OOD detection task. Thus, we have found **limited significant differences** when CoVer combined with these two methods, both in their high-level adapation and in our empirical results.
>
> **To guarantee the effectiveness of CoVer**, we recommend to use an additional validation set (e.g., the SVHN dataset used in our exploration) as the selection basis for choosing the effective corruption types and severity level, similar to previous works like Watermarking to learn a optimal mask using the validation set.

---

> > ### Comment · Reviewer_hcf7 · 2024-08-13
> >
> > Thank you for the feedback. I think the rebuttals do address my concern and would like to vote for accepting this paper.

---

> > > ### Author Response · Authors · 2024-08-14
> > > **Thanks for your positive feedback!**
> > >
> > > Thank you very much for the positive feedback! We are glad to hear that our response solved your concerns and will incorporate all suggestions in the revision.

---

### Official Review · Reviewer_NZqw · 2024-07-08

**Soundness:** 3
**Presentation:** 3
**Contribution:** 3
**Rating:** 7
**Confidence:** 4

**Summary:**

This paper first identifies the shortcoming of previous out-of-distribution (OOD) detection methods: the single-input paradigm limits the representation dimension for extracting valid information. To address this issue, a novel method CoVer that expands the original input space with multiple types of corruption is proposed. Based on the phenomenon of confidence mutation, CoVer achieves better performance by averaging the confidence scores across different dimensions. The proposed method achieves the best performance in the extensive experiments.

**Strengths:**

1. The motivation of the paper is easy to follow. The idea of using common corrupted images to expand the input space is both novel and reasonable. While previous methods based on single input often struggle with detecting hard-to-distinguish OOD samples, the proposed method offers a practical approach to improving the performance of OOD detection.

2. The extensive experiments demonstrates the solid contributions of the proposed work and significant improvements have ahieved for different methods. Using the ImageNet-1K benchmark, the experiments validate the compabability of the proposed CoVer method effectively.

3. The proposed method is designed with robust theoretical foundation of confidence mutation and the proposed score function. The analysis is presented clearly and accessibly for the novel insights, making it easy to understand the mechanism of the proposed framework.

**Weaknesses:**

1. Although the proposed method can improve the performance of the SOTA methods, whether they face the same challenges and how does the proposed method improve each method are not clearly explaiend.

2. For the different methods, what types of the inputs can improve them effectively lacks of deeply analysis.

**Questions:**

1. Since the proposed method is a framework to improve the OOD performance, I wonder how does the proposed CoVer method improve each method? Why the proposed method is effective to improve different types of the method should be further explaiend.
2. For the proposed method, it mainly focuses on expanding the inputs. While for the different method, they usually focus on different knowledge. I wonder how to decide what types of the inputs that can improve the performance of different methods.

**Limitations:**

Yes

---

> ### Author Rebuttal · Authors · 2024-08-07
>
> Thank you for your time devoted to reviewing this paper and your constructive suggestions. Here are our detailed replies to your questions.
>
> > **W1, Q1:** Although the proposed method can improve the performance of the SOTA methods, whether they face the same challenges and how does the proposed method improve each method are not clearly explained. Since the proposed method is a framework to improve the OOD performance, I wonder how does the proposed CoVer method improve each method? Why the proposed method is effective to improve different types of the method should be further explained.
>
> Thanks for your constructive comments. **First,** we would like to state that almost all the SOTA methods considered in our work face **the same challenge on excavating discriminative representation for OOD detection.** Although the SOTA methods are designed with different advanced techniques (e.g., ReAct, DICE, and ASH integrates the activation regularization or reshaping to the forward path of a single input in DNNs; MCM, LoCoOp, CLIPN, and NegLabel explore the CLIP's representation of a single input to detect OOD samples in VLMs), we notice that **the single input may implicitly constrain the representation dimension** for detection as the discriminative features are not infinite to utilize.
>
> **Second,** our CoVer introduce the extra dimensions with corruptions to **reveal the intrinsic distinguishes of ID and OOD samples.**  In the multi-inputs space, ID data maintains an overall higher confidence expectation, whereas OOD data encounters notable changes in model confidence since the high-frequency features are altered as verified in our analysis. Leveraging the difference trends between ID and OOD samples by a simple but effective averaging operation, our CoVer can further improve the ID-OOD separability when combined with those SOTA methods (i.e., adding the corrupted input dimension for scores averaging).
>
> > **W2, Q2:** For the different methods, what types of the inputs can improve them effectively lacks of deeply analysis. For the proposed method, it mainly focuses on expanding the inputs. While for the different method, they usually focus on different knowledge. I wonder how to decide what types of the inputs that can improve the performance of different methods.
>
> Thank you for your thoughtful question. For the types of corrupted inputs and their corresponding severity levels, we have conducted some related explorations (**e.g.,  Tables 10 and 11, Figures 6 and 7**) in the Appendix of our original submission for performance references. We have noticed that some specific corruptions (**e.g., Brightness, Fog, Contrast, Motion Blur, Defocus Blur**) can generally improve the OOD detection performance. It can be found **those types provide corruptions on the non-semantic level of the input**, instead of damaging the semantic feature too much like the other types. It is also empirically verified in our previous analysis about confidence mutation, are induced by the high-frequency feature changes. However, to further efficiently **determine which corruption types to choose** for expanding the input dimension, it is recommended to rely on an additional validation set (e.g., the SVHN dataset used in our exploration) as the selection basis, similar to the hyperparameter selection process in previous works.

---

> > ### Comment · Reviewer_NZqw · 2024-08-13
> >
> > Thank you for the detailed response, which addresses my concerns. After reviewing other comments, I still believe this paper is technically solid and novel. Therefore, I maintain my score.

---

> > > ### Author Response · Authors · 2024-08-13
> > > **Thanks for your positive support!**
> > >
> > > Thank you very much for your positive support after reading our response! We are glad to hear that our response addressed your concerns and that you find our paper technically solid and novel. We will make sure to incorporate all of your suggestions in the revision.

---

### Official Review · Reviewer_HCvU · 2024-07-08

**Soundness:** 3
**Presentation:** 3
**Contribution:** 3
**Rating:** 5
**Confidence:** 3

**Summary:**

Authors ntroduce a new approach to out-of-distribution (OOD) detection by expanding input representation dimensions using common corruptions. Traditional methods focus on single-input representations, limiting their effectiveness. This work identifies "confidence mutation," where OOD samples' confidence levels drop significantly under corruptions, while in-distribution (ID) data maintains higher confidence due to resistant semantic features.

The proposed method, Confidence aVerage (CoVer), averages confidence scores from original and corrupted inputs to improve OOD detection. Extensive experiments show that CoVer enhances performance across various benchmarks. Key contributions include:

1. Expanding input representation dimensions for better OOD detection.
2. Introducing confidence mutation to distinguish OOD data.
3. Proposing the CoVer scoring method for improved separability.
4. Validating CoVer's effectiveness through extensive experiments

This is a nice tricks that are many connection to others fields such a s NLP.

**Strengths:**

Nice idea which have not been applied to OOD detection but is well known on other fields such as NLP (e.g . retrieval)

**Weaknesses:**

**Omission of Data Depths and Information Projections in Related Work**

Data depths and information projections have shown significant promise in the OOD detection community due to their ability to provide robust and high-dimensional representations of data. Data depths, in particular, are well-suited for this problem as they project in all possible directions, capturing the entire distribution's structure and providing a natural fit for OOD detection tasks. Despite their relevance, these approaches are notably absent from the related work section of this paper. Including them could provide a more comprehensive overview of the field and strengthen the contextual foundation of the study. Relevant works that should be considered include:

M. Darrin. "Unsupervised Layer-wise Score Aggregation for Textual OOD Detection."

M. Picot. "A Halfspace-Mass Depth-Based Method for Adversarial Attack Detection." TMLR 2023.

P. Colombo. "Beyond Mahalanobis Distance for Textual OOD Detection." NeurIPS 2022.

**Questions:**

1. Why were data depths, which project onto all possible directions and are naturally suited for OOD detection, not explored in your related work?
2. Why were classical methods like Isolation Forest, which are well-known for their anomaly detection capabilities, not considered?
3. Can you compare your results against the previously introduced baselines involving data depths, information projections, and classical methods like Isolation Forest?

To me the paper is incomplete and the positioning as well as the choice of the methods (react, dice) is poor and deserve the paper, whereas classical ML tools are appropriated.

I encourage the authors to revise.

**Limitations:**

See above

---

> ### Author Rebuttal · Authors · 2024-08-07
>
> Thank you for your time devoted to reviewing this paper and your constructive suggestions. Here are our detailed replies to your questions.
>
> > **W1:** Omission of Data Depths and Information Projections in Related Work
>
> Thanks for your valuable suggestions and bringing us those insightful works of data depths and information projections. Here, **we have concluded a related discussion about these studies** in the following.
>
> Computing OOD scores on the embedding output of the last layer of the encoder is not the best choice for textual OOD detection. To address this, [1] proposed aggregating OOD scores across all layers and introduced an extended text OOD classification benchmark, MILTOOD-C. In a similar vein, RainProof [2] introduced a relative information projection framework and a new benchmark called LOFTER on text generators, considering both OOD performance and task-specific metrics. Building on the idea of information projection, REFEREE [3] leveraged I-projection to extract relevant information from the softmax outputs of a network for black-box adversarial attack detection. On the other hand, APPROVED [4] proposed to compute a similarity score between an input sample and the training distribution using the statistical notion of data depth at the logit layer. HAMPER [5] introduced a method to detect adversarial examples by utilizing the concept of data depths, particularly the halfspace-mass (HM) depth, known for its attractive properties and non-differentiability. Furthermore, TRUSTED [6] relied on the information available across all hidden layers of a network, leveraging a novel similarity score based on the Integrate Rank-Weighted depth for textual OOD detection. LAROUSSE [7] employed a new anomaly score built on the HM depth to detect textual adversarial attacks in an unsupervised manner.
>
> **We will add the above to the related work section for providing a comprehensive overview, and also add the formal citations for reference in our revised version.**
>
> > **Q1:** About data depths
>
> Thanks for your question. **We have discussed the related work on data depths in the previous response and will add it to the related work section in our revised version**. Since the discussion of the related work in our original submission mainly refers to recent representative reviews of visual OOD detection, like [8] and [9], we conduct limited exploration on this direction. Although our CoVer has conceptual distinguishes with exploring data depths for OOD detection, it is notable that **both provide new perspective for exploring discrimnative features** between the ID and OOD samples beyond the raw inputs.
>
> > **Q2:** About Isolation Forest
>
> Thanks for the question. We have noticed that Isolation Forest, though a classical method of anomaly detection (AD), is not a common baseline considered in previous literatures [8, 9] for OOD detection. **In fact, AD is quite different from OOD detection, as AD treats ID samples as a whole [9]**. This means that regardless of the number of classes (or statistical modalities) in ID data, AD does not require differentiation within the ID samples (while OOD detection does). OOD detection considers the knowledge for all the known classes, while AD mainly learns the normal patterns from the majority of data and identifies the anomalies. In our work, **we select the baseline methods following previous well-recognized studies**, such as DNN-based methods (e.g., ReAct, ASH, DICE) and VLM-based methods (e.g., MCM, CLIPN, NegLabel).
>
> > **Q3:** Comparison with data depths, information projections, and Isolation Forest
>
> Thank you for the question. **We have conducted comparison experiments between our CoVer and baselines as you mentioned**, as detailed **in Table 8 in attached PDF**.
>
> Due to the large scale of the ImageNet training set, we sampled 50 samples per class to construct a subset from the training data to represent the training distribution, as recommended by the similar work named NNGuide [10]. For data depths, we reimplemented APPROVED [4] for comparison. For information projections, we reproduced REFEREE [3] for comparison. For Isolation Forest, we use logits as the input to detect the anomaly logits in ID and OOD samples.
>
> The results indicate that AD and textual OOD detection methods, such as Data Depth and Information Projection, may not suit for visual OOD detection tasks, a view also mentioned in related surveys [8, 9]. Similarly, classical ML methods for AD, such as Isolation Forest, seem to be failed to excavate discriminative representations when applied to image OOD detection. However, since these methods are insightful in distinguishing the outliers, we believe **it is worth further efforts in the future to adopt the critical intuition into the OOD detection problem.**
>
> **References:**
>
>  [1] Darrin M, Staerman G, Gomes E D C, et al. Unsupervised Layer-wise Score Aggregation for Textual OOD Detection. In AAAI, 2024.
>
> [2] Darrin M, Piantanida P, Colombo P. ainproof: An Umbrella To Shield Text Generators From Out-Of-Distribution Data. In Arxiv, 2022.
>
> [3] Picot M, Noiry N, Piantanida P, et al. Adversarial Attack Detection Under Realistic Constraints. 2022.
>
> [4] Picot M, Staerman G, Granese F, et al. A Simple Unsupervised Data Depth-based Method to Detect Adversarial Images. 2023.
>
> [5] Colombo P, Picot M, Granese F, et al.  A Halfspace-Mass Depth-Based Method for Adversarial Attack Detection. TMLR, 2023.
>
> [6] Colombo P, Dadalto E, Staerman G, et al. Beyond Mahalanobis Distance for Textual OOD Detection. In NeurIPS, 2022.
>
> [7] Colombo P, Picot M, Noiry N, et al. Toward stronger textual attack detectors. In Arxiv, 2023.
>
> [8] Yang J, Wang P, Zou D, et al. Openood: Benchmarking generalized out-of-distribution detection. In NeurIPS, 2022.
>
> [9] Yang J, Zhou K, Li Y, et al. Generalized out-of-distribution detection: A survey. In IJCV, 2024.
>
> [10] Park J, Jung Y G, Teoh A B J. Nearest neighbor guidance for out-of-distribution detection. In ICCV, 2023.

---

> ### Author Response · Authors · 2024-08-10
> **Would you mind checking our responses and confirming whether you have any further questions?**
>
> Dear Reviewer HCvU,
>
> Thanks very much for your time and valuable comments on our work.
>
> In the rebuttal, we have tried our best to address the concerns, and provided detailed responses to all your comments and questions. Would you mind checking our responses and confirming if there is any unclear point so that we could further clarify?
>
> Best regards,
>
> Authors of Submission 1367

---

> ### Author Response · Authors · 2024-08-10
> **[Invitation to discussion] Need further clarification?**
>
> Dear Reviewer HCvU,
>
> Thanks very much for your time and valuable comments.
>
> Thanks again for your time and valuable comments. We have tried our best to address the concerns. Specifically, we
>
> - conclude a related discussion about the provided studies and cite them in our manuscript. (W1)
> - discuss data depths in our related work and add it to our manuscript. (Q1)
> - discuss isolation forest and clarify the difference between anomaly detection and OOD detection. (Q2)
> - verify the performance of CoVer with extra comparison experiments with the introduced baselines. (Q3)
>
> Is there any unclear point so that we should/could further clarify?
>
> Thanks for your attention and best regards,
>
> Authors of Submission 1367

---

### Official Review · Reviewer_LFoY · 2024-07-12

**Soundness:** 3
**Presentation:** 3
**Contribution:** 2
**Rating:** 4
**Confidence:** 5

**Summary:**

The paper aims to identify Out-Of-Distribution (OOD) samples by applying common image corruptions (noise, blur, etc.) to the input. The  phenomenon is referred to as confidence mutation, where original inputs, along with corruptions, increase the confidence of in-distribution (ID) data, while the confidence of OOD data decreases. Also, a new scoring function is proposed, Confidence aVerage (CoVer), which averages the OOD scores of different corrupted inputs and the original one.

Experiments are on 2 benchmarks (1) traditional OOD - ImageNet-1K as In-Distribution (ID) and iNaturalist, SUN, Places, and Textures as OOD (2) zero-shot OOD - mixture of ImageNet-10, ImageNet-20, ImageNet-100, WaterBirds, and Spurious. The metrics used for evaluation are AUROC and FPR95.

Methods exhibit better performance with the addition of CoVer.

**Strengths:**

The use of corruptions to improve OOD performance is creative. The implementation seems straightforward and does not require extra datasets. Also, CoVer seems to improve every method it's applied to.

The experiments utilize similar backbones/datasets as in the literature. This is helpful when comparing with other methods.

All figures in the paper, including Fig. 1-4, are intuitive and easy to follow. The captions are detailed, and the overview (Fig. 4) makes understanding CoVer very straightforward.

The paper is well-written and clear. It effectively discusses the problem, presents the idea, and describes the experiments in a coherent manner. Additionally, all equations are clearly explained.

**Weaknesses:**

The experimentation/idea is somewhat weak and seems more like an application rather than significant knowledge advancement. For example, CoVer is not standalone but applied to an existing method like ASH (see Table 1) or on top of a VLM, thereby leveraging the performance of other methods. Additionally, the claimed contribution of a novel scoring method doesn't seem entirely novel, as averaging is straightforward and an existing scoring function like Softmax/Energy is subsequently employed.

Comparisons in Table 1 with different baselines is a little unfair because CoVer makes use of "extra" data (corrupted images), while others simply use a single input/image.

Evaluating OOD performance when using VLM/CLIP models is problematic. Such models utilize a vast amount of data which makes it near impossible to know what images/classes have been seen before and are considered "in-distribution". While the community continues to use them in various settings, their use with OOD problems should be a clear limitation.

I find it unclear how to determine which corruption types or severity levels to use if I implemented CoVer. The results suggests variability/inconsistency without a clear guideline for decision-making.

The runtime of CoVer is not discussed. From my understanding, one would need to pass every corrupted input and extract its features, which is time consuming and uses more space.

There is no discussion of failure cases. When using corrupted inputs, I assume there are instances when CoVer labeled an ID image as OOD.

Since the OOD datasets used in Table 1 differ significantly from ImageNet, it would be intriguing to observe CoVer's performance on more challenging ones such as NINCO [1] (ensures no categorical contamination) without leveraging CLIP.

The experiments do not utilize or compare with NNGuide [2] or MaxLogit [3], more recent and better performing methods than those in the paper.

[1] Bitterwolf, Julian, Maximilian Müller, and Matthias Hein. "In or Out? Fixing ImageNet Out-of-Distribution Detection Evaluation." International Conference on Machine Learning. PMLR, 2023.

[2] Park, Jaewoo, Yoon Gyo Jung, and Andrew Beng Jin Teoh. "Nearest Neighbor Guidance for Out-of-Distribution Detection." Proceedings of the IEEE/CVF International Conference on Computer Vision. 2023.

[3] Hendrycks, Dan, et al. "Scaling Out-of-Distribution Detection for Real-World Settings." International Conference on Machine Learning. PMLR, 2022.

**Questions:**

In Table 1, why is CoVer only combined with ASH and not other methods?

Table 6 in the appendix demonstrates that CoVer is integrated with other methods using different types of corruptions and severity levels for each method. This raises questions about the comparisons. Why are the types and levels of corruptions not the same across all methods?

Since the paper has different ablations (various corruption types, number of dimensions/corrupted inputs, severity levels) and the experiments make an explicit choice on each one, was a validation set ever used? I did not see anything in the code or the paper, indicating choices were tuned on the test set.

Considering that other methods only use one input, how much additional runtime does CoVer add? For example, given the size of ImageNet-1K, using CoVer would necessitate extracting 6 million features (1.2 million * 5 corruptions).

How effective would CoVer be if it encountered much more difficult OOD data without leveraging VLMs/CLIP?

**Limitations:**

The authors discussed limitations and broader impact in Section Appendix E.

---

> ### Author Rebuttal · Authors · 2024-08-07
>
> Thank you for your time devoted to reviewing this paper and your constructive suggestions. Here are our detailed replies to your questions.
>
> >**W1:** About Novelty
>
> We would like to reclaim that our work is novel in introducing a new perspective, i.e., expanding the dimension of input with corrupted variants, for OOD detection. **This has not been explored in previous methods [1, 2]**, and has both **conceptual and techniqual distinguishes** from them with significant knowledge advancement.
>
> **Conceptually,** we are the first to expand the raw input with the corrupted ones and identify the phenomenon of confidence mutation, and reveal the underlying mechanism with high-frequency feature change. **The multiple input dimensions introduced in our framework** is the critical advance compared with previous ones focusing on single-input.
>
> **Techniqually,** CoVer captures the intrinsic differences of ID and OOD samples using the averaging operation, as founded to be **simple yet highly effective** and evidenced by our experiments in **Table 1,2,3 of the original submission**. As our idea provides a different perspective, the combination with these SOTA methods aims to demonstrating that **our CoVer reveals the additional enhancement not covered by the basis**. Similar to previous methods (ASH, DICE, ReAct) originate from Softmax/Energy, we believe our work brings new insights on this problem.
>
> > **W2:** About "extra data"
>
> Thank you for the question. We would like to clarify that **it is our novelty to use extra corrupted inputs to expand the input space, and it is not unfair.** While previous OOD detection methods have distinguished between ID and OOD data based on a single input, **we are the first to identify the challenges in this paradigm and propose expanding it to multiple inputs**, measuring their confidences respectively. With these novel insights, CoVer **provides a new perspective** for the OOD detection problem.
>
> > **W3:** About using VLMs/CLIP
>
> Thank you for the insightful comments.
>
> First, we have conducted various experiments of CoVer on the **DNNs architecture (ResNet50)**. The results demonstrate performance improvements and **exhibit the same trend as those on VLMs** (**refer to Table 1 and Table 3 of the original submission**).
>
> Second, we would like to explain that **the zero-shot settings in our work follow the MCM [3]**, which is the pioneering work in zero-shot OOD detection with VLM. As in MCM, the in-distribution classes in zero-shot OOD detection are the classification task of interest, which is defined by a set of class labels/names $Y_{in}$ instead of the classes used in pre-training. Accordingly, OOD is defined w.r.t. the ID classes, not the data distribution during pre-training. Hence, we only utilize the powerful image encoder to extract visual features and the text encoder to extract textual embeddings as ID concept prototypes, without the prior knowledge for distinguishing ID and OOD classes. We will add this part of explanation to our paper.
>
> > **W4, Q2:** About which corruption types or severity levels to use
>
> Thanks for your constructive comments.
>
> First, in all experiments, we use the **SVHN dataset as the validation set** to select the most effective corruption types for each method. **We will clearly explain the usage of the validation set in our main paper**.
>
> For the types of corrupted inputs and their corresponding severity levels, we have conducted related explorations (**e.g.,  Tables 10 and 11, Figures 6 and 7 of our original submission**) for performance references. Some specific corruptions (**e.g., Brightness, Fog, Contrast, Motion Blur, Defocus Blur)** can generally improve the OOD detection performance, as the corruptions are mainly on the non-semantic level of the input, instead of damaging the semantic features too much like the other types.
>
> Empirically, refer to **Table 3 in attached PDF**, we can use **the same type of corruption** as the expanded input (e.g., here Brightness with severity 1 used) to perform better than the original version. This provides the verification of the previous intuition about the general guidance for choosing appropriate corruption types, and understanding dimension expansion for OOD detection.
>
> > **W5, Q4:** About runtime
>
> Thank you for your valuable question. Runtime is indeed an issue we didn't consider. As you mentioned, if there are $N$ expanded dimensions, it will take $N$​ times the duration of a single input to implement CoVer. However, **our CoVer is only applied in the inference phase of OOD detection**, and it is **generally fast**. As shown in **Table 4 in attached PDF**, we report the inference time of each single input on ID and OOD datasets. **We will clearly discuss the runtime issue in the revised version.**
>
> > **W6:** About failure cases
>
> Thank you for the constructive comment. We indeed identified some failure cases and reported them in **Figure 7 of the original submission**. When CoVer utilizes certain severe corruption types (e.g., Spatter, Elastic transform), its performance is worse than with single input. This is because these types are more severe compared to others, leading to **excessive damages to semantic features**. Effective corruption types are those **only perturb non-semantic features**, which generally exist at the high-frequency level, resulting in different confidence variations between ID and OOD data. We will incorporate these discussions into the revised version and make it clearer with our empirical results.
>
> **References**:
>
> [1] Yang J, Wang P, Zou D, et al. Openood: Benchmarking generalized out-of-distribution detection. In NeurIPS, 2022.
>
> [2] Yang J, Zhou K, Li Y, et al. Generalized out-of-distribution detection: A survey. In IJCV, 2024.
>
> [3] Ming Y, Cai Z, Gu J, et al. Delving into out-of-distribution detection with vision-language representations. In NeurIPS, 2022.
>
> > **W7, Q5, W8, Q1, Q3**
>
> Due to the space limit, we place our answer in the general response.

---

> ### Author Response · Authors · 2024-08-08
> **Would you mind checking our responses and confirming whether you have any further questions?**
>
> Dear Reviewer LFoY,
>
> Thanks very much for your time and valuable comments on our work.
>
> In the rebuttal, we have tried our best to address the concerns, and provided detailed responses to all your comments and questions. Would you mind checking our responses and confirming if there is any unclear point so that we could further clarify?
>
> Best regards,
>
> Authors of Submission 1367

---

> ### Author Response · Authors · 2024-08-10
> **[Invitation to discussion] Need further clarification?**
>
> Dear Reviewer LFoY,
>
> Thanks again for your time and valuable comments. We have tried our best to address the concerns. Specifically, we
>
> - reclaim our novelty from both conceptual and technical perspectives (W1);
> - clarify the fair comparison in using CoVer's novel expanded inputs. (W2);
> - discuss the rationale for using VLM/CLIP models in OOD detection experimentally and definitionally. (W3);
> - clarify the usage of the validation set and provide general guidance for choosing appropriate corruption types empirically. (W4, Q2, Q3)
> - discuss the runtime issue with extra experimental results. (W5)
> - further clarify the failure cases in practice. (W6)
> - verify the performance of CoVer on NINCO dataset with extra experiments. (W7, Q5)
> - verify the performance of CoVer with comparison with NNGuide and MaxLogit methods. (W8)
> - explain the reason for combining only with ASH and demonstrate the effectiveness of combining other methods empirically. (Q1)
>
> Is there any unclear point so that we should/could further clarify?
>
> Thanks for your attention and best regards,
>
> Authors of Submission 1367

---

> > ### Comment · Reviewer_HCvU · 2024-08-10
> >
> > Thank you for your response. While I appreciate the enthusiasm of authors for feedback, I have other commitments in addition to the NeurIPS reviews. I will get back to you as soon as possible, and I kindly ask for your patience in the meantime.

---

> ### Comment · Reviewer_LFoY · 2024-08-11
> **Rebuttal Acknowledgement**
>
> I thank the authors for taking time to submit a rebuttal and providing some clarity.
>
> > General Response (NINCO, NNGuide and MaxLogit, ASH)
>
> Thanks for the comparison with more recent data and methods.
>
> Using SVHN as a validation set is an interesting choice because the data is much smaller (32x32) than ImageNet (ID set) or any of the OOD datasets.
>
> > Knowledge Advancement
>
> My comment should have indicated that the advancement in knowledge is not as significant, rather than suggesting that the idea itself is weak. CoVer requires [1] (corruptions and severity levels) and must be applied to an existing OOD method. Other work [2] has utilized [1] in OOD detection, but simply as an attack. The contribution in terms of knowledge advancement or new perspective is using a normal image in conjunction with a corrupted image/s for post-hoc OOD detection enhancement by averages scores.
>
> Utilizing CoVer would require applying it to every method for a fair comparison rather than using it as a standalone technique, as it involves more than a single input. This isn't necessarily bad or wrong, but it is an important consideration when determining its impact. Its application does not alter performance rankings, and one would generally expect performance to improve with the use of multiple inputs.
>
> > Extra data
>
> I believe this should be listed as a limitation because, for CoVer to be effective, it requires more than one input compared to other methods. Similar to how OpenOOD distinguishes between "Training Methods" and "Training With Extra Data," CoVer would need to be categorized as "Post-hoc Methods With Extra Data." This is why I mentioned that Table 1 seemed a bit unfair, as it compares methods using one input vs. two or more (CoVer).
>
> > VLMs/CLIP
>
> I am aware how MCM and OOD are defined, but my point is an image encoder coming from a VLM/CLIP model is of course going to be better because it has seen more data than one with only ImageNet-1K. It is an advantage that should be noted.
>
> > Corruption types / severity levels
>
> The variability I mentioned arises because the experiments used different corruption types and/or severity levels for nearly every method. This makes comparisons less discernible. Also, it introduces variability depending on the architecture, dataset, and other factors.
>
> The attached PDF demonstrates that using a single corruption type along with a normal input improves AUROC and FPR95 alone.
>
> > Runtime
>
> Thank you for attaching the runtime table. My concern is that if one were to use CoVer and explore corruption types and severity levels across different architectures and their own data, it could become time-consuming due to the amount of variability involved.
>
> > Failure cases
>
> This was my oversight, as I missed Figure 7. I was simply wondering in what situations the use of CoVer might not make sense and a single input would be preferred.
>
>
> **References**:
>
> [1] Hendrycks, D., & Dietterich, T. (2018). Benchmarking Neural Network Robustness to Common Corruptions and Perturbations. In International Conference on Learning Representations.
>
> [2] Chen, J., Li, Y., Wu, X., Liang, Y., & Jha, S. (2021). ATOM: Robustifying Out-of-distribution Detection Using Outlier Mining. In Machine Learning and Knowledge Discovery in Databases.

---

> > ### Author Response · Authors · 2024-08-12
> > **[1/2] Thanks for your feedback!**
> >
> > We appreciate the reviewer's engagement during the discussion phase. Below are our detailed response to your comments.
> >
> > > About SVHN as the validation set
> >
> > Thanks for acknowledging our results on advanced comparison.
> >
> > We would like to clarify that it is necessary to **sclae the size** of SVHN data from **32 $\times$ 32 to 224 $\times$ 224** when using the SVHN dataset as the validation set on the ImageNet benchmark, as same to the usage in previous work like Watermarking [1]. This constraint ensures that the size of SVHN data can be consistent with the data size of ImageNet and other OOD datasets.
> >
> > > About knowledge advancement
> >
> > Thanks for the clarification on your concern.
> >
> > First, **regarding the knowledge advancement**, we believe it is significant. While it is true that previous works used corruptions in OOD detection, e.g., as an attack [2], the key difference here lies in **our innovative use of normal and corrupted images in conjunction to enhance post-hoc OOD detection through average scores.** Futhermore, considering corruptions as attacks in [2], it also **needs to be applied within an existing OE framework and utilize the original OOD score**, same as our CoVer also requires integration with existing OOD scoring function, but with a distinct purpose and perspective.
> >
> > Second, **concerning the impact and fair comparison**, we would like to clarify that **many other methods also need to be integrated with existing techniques and OOD scores** (e.g., DICE, ReAct, ASH in our previous response). Our approach, to combine with other OOD scoring functions and methods, **is intended to demonstrate the validity of our new perspective.** While it's true that utilizing multiple inputs could generally improve performance, the primary contribution of our work is the introduction of a novel approach that **offers a fresh insight on post-hoc OOD detection**.
> >
> > > About extra data
> >
> > Thanks for the suggestion.
> >
> > First, we would like to conceptually reclarify that the corrupted forms of existing images is actually not the "extra data", since **extra data is generally considered as those that do not intersect with the existing samples in their semantic label spaces [3]**. For instance, outlier data used in outlier exposure (OE) based methods is regarded as "extra data", since they belongs to disjoint label spaces. However, our expanded inputs originate from the existing data, which shares the same label space but are corrupted.
> >
> > Second, we would like to reclaim that our innovation lies in expanding the input with corruption, as highlighted in the title of our research. **Even if using two or more** original inputs **without corruptions**, these methods would yield similar results to those obtained with a single input, underscoring the importance of our approach. Our CoVer sheds the light of using corruptions to effectively expand the input, which sets us apart.
> >
> > Nevertheless, we acknowledge this in our limitations, since our CoVer does introduce multiple inputs, which may increase computational costs. **However, this should not be viewed as conceptually unfair, as the advantage lies in the new perspective of our method rather than an inherent imbalance.** **For instance,** NegLabel [4] proposed introducing numerous negative words into CLIP (which is also like "extra data") to boost the performance of post-hoc OOD detection without labeling their method as 'Post-hoc Methods With Extra Data' or comparing it with other methods also expanded with negative labels. **This further highlights the importance of the new perspective compared with the fairness issue, similar to our current case.**
> >
> > > About VLMs/CLIP
> >
> > Thank you for your valuable suggestion. **We will clearly note the advantage of VLMs/CLIP used for OOD detection in the revised version.**
> >
> > An additional note is that it is because CLIP's image encoder has seen more data and is more powerful that CLIP can be used for zero-shot OOD detection without any additional data for training.
> >
> > > About corruption types / severity levels
> >
> > Thanks for your acknowledgement!
> >
> > As you mentioned, the lack of a clear guideline on how to select the proper types of corruption will result in negligible variability of CoVer in practice. Following your suggestion, **we will merge all the results into our manuscript with detailed introduction and discussion of the clear guideline on appropriate corruption types.**
> >
> > **References:**
> >
> > [1] Wang Q, Liu F, Zhang Y, et al. Watermarking for out-of-distribution detection. In NeurIPS, 2022.
> >
> > [2] Chen J, Li Y, Wu X, et al. Atom: Robustifying out-of-distribution detection using outlier mining. In Machine Learning and Knowledge Discovery in Databases, 2021.
> >
> > [3] Yang J, Zhou K, Li Y, et al. Generalized out-of-distribution detection: A survey. In IJCV, 2024.
> >
> > [4] Jiang X, Liu F, Fang Z, et al. Negative label guided ood detection with pretrained vision-language models. In ICLR, 2024.

---

> > ### Author Response · Authors · 2024-08-12
> > **[2/2] Thanks for your feedback!**
> >
> > > About runtime
> >
> > Thank you for your constructive comment.
> >
> > **Technically,** we have provided the implementations of different corruptions in the submitted code, and **we also provided exploration on different corruption types and severity which can provide some general insights on choosing  the corruption types in our previous response.** If someone were to use CoVer on different architectures and their own data, they could simply construct their own corrupted datasets and refer to the provided general guideline to select the appropriate corruption types. Additionally, we recommend to use the validation set for the performance tuning.
> >
> > **As for the variability,** we have conducted various experiments on both mainstream and challenging OOD datasets (including iNaturalist, SUN, Places, Textures, NINCO), the results have all demonstrated that the provided general guideline (e.g., single expanded input using Brightness with severity 1) is always the better choice for implementing CoVer. As a result, **the variability that exists in implementing CoVer can be effectively minimized when a general guideline is available.**
> >
> > As the common issue for those advanced scoring method design (e.g., DICE, ReAct, ASH in our previous response) introducing the manipulation on feature or parameter level adjustment, **we would also discuss it as one part of limitation in the revised version.**
> >
> > > About failure cases
> >
> > Thanks for the clarification.
> >
> > As in our previous response, certain corruption types that **excessive damage to semantic features of images** would result in CoVer perform worse than the single input. This further suggests a potential guidance in choosing effective types of corruption which **perturb the non-semantic features**. **We will highlight this part in main text as to provide a comprehensive study for readers on this aspect.**
> >
> > **Thanks again for your acknowledgement and the response in improving our work, we are happy to discuss further if there is any questions and remaining concerns.**

---

### Author Rebuttal · Authors · 2024-08-07

## General Response

We appreciate all the reviewers for their thoughtful comments and suggestions on our paper.

We are very glad to see that the reviewers find **our focused problem is important** (R3,R4) within the OOD detection research, and the method is **novel** (R1,R2,R3,R4) and **simple but adaptable** (R1,R2,R3,R4) to various other techniques, and the **experiments are good, comprehensive** and demonstrate the **general effectiveness** of our CoVer (R1,R2,R3,R4). We are also pleased that the reviewers find our writing and figures is **very clear** and **easy to understand** (R1,R4).

We have tried our best to address the reviewers' comments and concerns in **individual responses to each reviewer** with comprehensive experimental justification. The reviews allowed us to improve our draft and the contents added in the revised version and **the attached PDF** are summarized below:

**From Reviewer LFoY**

- Clarify and Discuss our main novelty and settings of CoVer. (see Section 1,  3.1, and 3.2 in the original submission, will highlight in these sections)
- Summarize and add results for expanded types and runtime issues (see Table 3,4 in PDF, will add in Appendix F)
- Discuss the failure cases of CoVer (will add and highlight in the limitation part of Appendix E)
- Add the experimental verification on NINCO dataset and compare with NNGuide and MaxLogit (see Table 1,2,5 in PDF, will add in Appendix F)
- Conduct comparison experiments on each mentioned DNN-based methods (see Table 6 in PDF, will add in Appendix F)
- Clarify and Explain the usage of validation set in CoVer (see Table 7 in PDF, will add and highlight in Section 4.1).

**From Reviewer HCvU**

- Discuss and add the reference for methods in related fields. (will add in Appendix B)

- Explain the differences of data depths, information projections and Isolation Forest,  and their challenges in visual OOD detection. (will add in Appendix F)


- Add the experimental comparison with newly considered methods. (see Table 8 in PDF, will add in Appendix F)

**From Reviewer NZqw**

- Discuss and revise the statement for our critical observations and add more explanation (will revise in Section 1 and 3.1)

- Show our results about the effects of expanded types and provide some insights on choosing corruption types. (will add and highlight in Section 4.1)


**From Reviewer hcf7**

- Add the experimental comparison with Watermarking. (see Table 9 in PDF, will add in Appendix F)

- Discuss the generalization and effectiveness of CoVer. (see Section 3.1 in the original submission)

**We appreciate your comments and time!** We have tried our best to address your concerns and revised the paper following the suggestions. **Would you mind checking it and confirming if there are any unclear parts?**

---

### Some rest answers:

For **Reviewer LFoY**

> **W7, Q5:** Experiments on NINCO

Thank you for your valuable question. NINCO has proposed three OOD datasets with no categorical contamination which include NINCO, OOD unit-tests, and NINCO popular OOD datasets subsamples. Here, we evaluate the effectiveness of CoVer on these datasets in **Table 5 in attached PDF**. The results demonstrate that CoVer, when combined with ASH, consistently achieves better performance across the three NINCO OOD datasets.

> **W8:** Comparison with NNGuide and MaxLogit

Thanks for your valuable suggestion. We have conducted comparison experiments with NNGuide and MaxLogit to enrich our analysis in **Table 1 in attached PDF**.

First, our experimental results show that CoVer outperforms these competitive post-hoc methods on the ResNet50 architecture.

Second, the performance of these post-doc methods, especially NNGuide, **encounter significant drop when conducted on CLIP-B/16 architecture**. We believe the reason for the poor performance is **the difference in training data**. Many pos-hoc methods are designed on ImageNet pre-trained networks, where only ID data are used during training. In contrast, when training CLIP, both ID datas and OOD datas are used. This leads to different activations of OOD data. Another reason is that the pos-hoc method relies heavily on **the choice of hyperparameters**. The hyperparameters of NNGuide need to be re-selected on different models. Despite these issues, our CoVer can still perform better than these methods.

Furthermore, we also **combine our CoVer with MaxLogit and NNGuide** and report the results in **Table 2 in attached PDF**, which further demonstrates the effectiveness and compatibility of our method.

> **Q1:** why only combined with ASH, not the others?

Thank you for your question. In Table 1 in the original submission, we only reported the results of CoVer combined with ASH **because it best demonstrates the excellence of CoVer**. In **Table 3 in the original submission**, we also show the results of CoVer combined with DICE and ReAct, and CoVer can also provide performance gains for them. Here in **Table 6 in attached PDF, we further report the comparison of CoVer combined with each mentioned DNN-based methods** (adding MSP, ODIN, and Energy score), which strongly demonstrates its superiority.

> **Q3:** About validation set

Thank you for your insightful questions. We utilized **the SVHN dataset as the validation set** to determine the most effective corruption types for each method in all experiments. Specific examples of selections are provided in **Table 7 in attached PDF. We will clearly explain the usage of the validation set in our main paper**. We would like to explain that our submitted code is developed from the MCM code repository, and methods such as MCM and NegLabel did not explicitly employ a validation set in their implementation. Furthermore, the main function realization of the submitted code focuses on the critical implementation steps of CoVer for the usage, omitting the validation set part. **We will definitely add relevant code in the revised version and clearly explain it**.

---

### Decision · Program_Chairs · 2024-09-25

**Decision:**

Accept (poster)

**Comment:**

After reading the paper and the reviews, with significant reviewer discussion, I am voting to accept the paper. The use of multiple realizations of the input via random transformations being used at once is a bit counter-intuitive but has been shown to provide a good lift to improve OOD detection, and is applicable to multiple different OOD detectors. A large number of datasets are tested with different models, showing consistent improvement in most cases. While the AUC improvements are modest, the FPR gains are larger - which implies that the nature of the improvement comes more from calibration than new capability. Though calibration can be done with a secondary set, in my view, there is a non-trivial value to calibration absent explicit secondary mechanisms like a calibration set as it simplifies development/deployment. While there is some related work with the same inspiration for adversarial attacks[1], the approach is more than sufficiently different. The largest detractor brought by some reviewers are concerns around novelty, and related work not included.  Data augmentations are not new, and the specific augmentations are not new - but the method of applying them is. Given that two reviewers voted to accept with high scores, I'm inclined to believe that more than one future reader will be interested in using/leveraging this paper and its approach. I don't see any reviewer concerns that the approach itself is invalid/had some evaluation flaw, but I think the content of this meta-review and the feedback given by the reviewers should be carefully included in the camera ready to ensure an appropriate level of claim is made. Specifically, the claims of being "the first to expand" and offering a "new perspective" should be tempered to the reality of the paper and shouldn't be a distraction from the otherwise good writing and interesting results. Several related works were noted that should also be included in the camera-ready version.

1. Barrage of Random Transforms for Adversarially Robust Defense. In The IEEE Conference on Computer Vision and Pattern Recognition (CVPR). 2019